# Ancient role of sulfakinin/cholecystokinin-type signalling in inhibitory regulation of feeding processes revealed in an echinoderm

Ana B Tinoco[1†], Antón Barreiro-Iglesias[1†‡], Luis Alfonso Yañez Guerra[1†§], Jérôme Delroisse[1#], Ya Zhang[1], Elizabeth F Gunner[1], Cleidiane G Zampronio[2], Alexandra M Jones[2], Michaela Egertová[1], Maurice R Elphick[1*]

[1]Queen Mary University of London, School of Biological & Behavioural Sciences, London, United Kingdom; [2]School of Life Sciences and Proteomics, Research Technology Platform, University of Warwick, Coventry, United Kingdom

*For correspondence:
m.r.elphick@qmul.ac.uk

†These authors contributed equally to this work

Present address: ‡Department of Functional Biology, CIBUS, Faculty of Biology, Universidade de Santiago de Compostela, Santiago de Compostela, Spain; §Living Systems Institute, University of Exeter, Exeter, United Kingdom; #University of Mons, Biology of Marine Organisms and Biomimetics Unit, Mons, Belgium

Competing interest: The authors declare that no competing interests exist.

**Abstract** Sulfakinin (SK)/cholecystokinin (CCK)-type neuropeptides regulate feeding and digestion in protostomes (e.g. insects) and chordates. Here, we characterised SK/CCK-type signalling for the first time in a non-chordate deuterostome – the starfish *Asterias rubens* (phylum Echinodermata). In this species, two neuropeptides (ArSK/CCK1, ArSK/CCK2) derived from the precursor protein ArSK/CCKP act as ligands for an SK/CCK-type receptor (ArSK/CCKR) and these peptides/proteins are expressed in the nervous system, digestive system, tube feet, and body wall. Furthermore, ArSK/CCK1 and ArSK/CCK2 cause dose-dependent contraction of cardiac stomach, tube foot, and apical muscle preparations in vitro, and injection of these neuropeptides in vivo triggers cardiac stomach retraction and inhibition of the onset of feeding in *A. rubens*. Thus, an evolutionarily ancient role of SK/CCK-type neuropeptides as inhibitory regulators of feeding-related processes in the Bilateria has been conserved in the unusual and unique context of the extra-oral feeding behaviour and pentaradial body plan of an echinoderm.

## Introduction

The peptide hormones cholecystokinin (CCK) and gastrin were discovered and named on account of their effects as stimulators of gall bladder contraction and gastric secretion of pepsin/acid, respectively, in mammals (*Edkins, 1906*; *Ivy ACO and Goldberg, 1928*). Determination of the structures of CCK and gastrin revealed that they have the same C-terminal structural motif (Trp-Met-Asp-Phe-NH₂), indicative of a common evolutionary origin (*Gregory et al., 1964*; *Mutt and Jorpes, 1968*). However, CCK and gastrin are derived from different precursor proteins, which are subject to cell/tissue-specific processing to give rise to bioactive peptides of varying length (e.g. CCK-8 and CCK-33; gastrin-17 and gastrin-34) (*Boel et al., 1983*; *Deschenes et al., 1984*; *Rehfeld et al., 2007*). Furthermore, CCK and gastrin have a tyrosine residue at positions 7 and 6 from the C-terminal amide, respectively, which can be sulphated post-translationally (*Rehfeld et al., 2007*). The effects of CCK and gastrin in mammals are mediated by two G-protein-coupled receptors (GPCRs), CCK-A (CCKR1) and CCK-B (CCKR2), with both sulphated and non-sulphated forms of CCK and gastrin acting as ligands for the CCK-B receptor, whilst the CCK-A receptor is selectively activated by sulphated CCK (*de Weerth et al., 1993*; *Dufresne et al., 2006*; *Kopin et al., 1992*; *Lee et al., 1993*; *Noble and Roques, 1999*; *Wank et al., 1992*). Mediated by these receptors, gastrin and CCK have a variety of physiological/behavioural effects in mammals. Thus, in the gastrointestinal system, gastrin stimulates growth of the

stomach lining, gastric contractions, and gastric emptying (*Crean et al., 1969*; *Dockray et al., 2005*; *Gregory and Tracy, 1964*; *Vizi et al., 1973*), whilst CCK stimulates pancreatic enzyme secretion, contraction of the pyloric sphincter, and intestinal motility (*Chen et al., 2004*; *Gutiérrez et al., 1974*; *Harper and Raper, 1943*; *Rehfeld, 2017*; *Shaw and Jones, 1978*; *Vizi et al., 1973*). Furthermore, CCK also has behavioural effects that include inhibition of food intake as a mediator of satiety and stimulation of aggression and anxiogenesis (*Chandra and Liddle, 2007*; *Gibbs et al., 1973*; *Singh et al., 1991*; *Smith et al., 1981*).

Phylogenomic studies indicate that genome duplication in a common ancestor of the vertebrates gave rise to genes encoding CCK-type and gastrin-type precursor proteins (*Dupré and Tostivint, 2014*). Accordingly, invertebrate chordates that are the closest extant relatives of vertebrates (e.g. the urochordate *Ciona intestinalis*) have a single gene encoding a 'hybrid' CCK/gastrin-like peptide (e.g. cionin) with a sulphated tyrosine residue at both positions 6 and 7 from the C-terminal amide (*Johnsen and Rehfeld, 1990*; *Monstein et al., 1993*; *Thorndyke and Dockray, 1986*). Furthermore, CCK-type peptides stimulate gastric enzyme secretion in the sea squirt *Styela clava*, providing evidence of evolutionarily ancient roles as regulators of gastrointestinal physiology in chordates (*Bevis and Thorndyke, 1981*; *Thorndyke and Bevis, 1984*).

Evidence that the phylogenetic distribution of CCK/gastrin-type peptides may extend beyond chordates to other phyla was first obtained with the detection of substances immunoreactive with antibodies to CCK and/or gastrin in a variety of invertebrates, including arthropods, annelids, molluscs, and cnidarians (*Dockray et al., 1981*; *El-Salhy et al., 1980*; *Grimmelikhuijzen et al., 1980*; *Kramer et al., 1977*; *Larson and Vigna, 1983*; *Rzasa et al., 1982*). However, molecular evidence of the evolutionary antiquity of CCK/gastrin-type signalling was obtained with the purification and sequencing of a CCK-like peptide named leucosulfakinin, which was isolated from the insect (cockroach) *Leucophaea maderae* (*Nachman et al., 1986a*). Subsequently, GPCRs that are homologs of the vertebrate CCKA/CCKB-type receptors have been identified and pharmacologically characterised as receptors for sulfakinin (SK)-type peptides in a variety of insects, including *Drosophila melanogaster* (*Bloom et al., 2019*; *Kubiak et al., 2002*; *Yu et al., 2013b*; *Yu and Smagghe, 2014b*). Furthermore, investigation of the physiological roles of SK-type signalling in insects has revealed similarities with findings from vertebrates. Thus, in several insect species SK-type peptides have myotropic effects on the gut (*Al-Alkawi et al., 2017*; *Marciniak et al., 2011*; *Nachman et al., 1986a*; *Nachman et al., 1986b*; *Nichols, 2007*; *Palmer et al., 2007*; *Predel et al., 2001*; *Schoofs et al., 1990*) and/or affect digestive enzyme release (*Harshini et al., 2002a*; *Harshini et al., 2002b*; *Nachman et al., 1997*; *Zels et al., 2015*). Furthermore, at a behavioural level there is evidence that SK-type peptides act as satiety factors (*Al-Alkawi et al., 2017*; *Bloom et al., 2019*; *Downer et al., 2007*; *Maestro et al., 2001*; *Meyering-Vos and Müller, 2007*; *Nässel and Zandawala, 2019*; *Nichols et al., 2008*; *Wei, 2000*; *Yu et al., 2013a*; *Yu et al., 2013b*; *Yu and Smagghe, 2014b*; *Zels et al., 2015*) and regulate locomotion and aggression in insects (*Chen et al., 2012*; *Nässel and Williams, 2014*; *Nässel and Zandawala, 2019*; *Nichols et al., 2008*).

The discovery and functional characterisation of SK-type signalling in insects and other arthropods indicated that the evolutionary origin SK/CCK-type signalling can be traced back to the common ancestor of the Bilateria. Consistent with this hypothesis, SK/CCK-type signalling systems have been discovered in a variety of protostome invertebrates, including the nematode *Caenorhabditis elegans*, the mollusc *Crassostrea gigas,* and the annelid *Capitella teleta* (*Janssen et al., 2008*; *Mirabeau and Joly, 2013*; *Schwartz et al., 2018*). Furthermore, some insights into the physiological roles of SK/CCK-type signalling in non-arthropod protostomes have been obtained, including causing a decrease in the frequency of spontaneous contractions of the *C. gigas* hindgut (*Schwartz et al., 2018*), stimulation of digestive enzyme secretion in *C. elegans* and *Pecten maximus* (*Nachman et al., 1997*) and evidence of a role in regulation of feeding and energy storage in *C. gigas* (*Schwartz et al., 2018*).

Little is known about SK/CCK-type signalling in the Ambulacraria (echinoderms and hemichordates) – deuterostome invertebrates that occupy an 'intermediate' phylogenetic position with respect to protostomes and chordates (*Furlong and Holland, 2002*; *Telford et al., 2015*). Prior to the genome sequencing era, use of immunohistochemical methods revealed CCK-like immunoreactive cells in the intestine of sea cucumbers (phylum Echinodermata) and vertebrate CCK/gastrin-type peptides were found to cause relaxation of sea cucumber intestine (*García-Arrarás et al., 1991*). More recently, analysis of transcriptome/genome sequence data has enabled identification of transcripts/genes encoding

SK/CCK-type peptide precursors and SK/CCK-type receptors in echinoderms and hemichordates (*Burke et al., 2006*; *Chen et al., 2019*; *Jékely, 2013*; *Mirabeau and Joly, 2013*; *Semmens et al., 2016*; *Zandawala et al., 2017*). However, functional characterisation of native SK/CCK-type peptides and receptors has yet to be reported for an echinoderm or hemichordate species. We have established the common European starfish *Asterias rubens* as an experimental model for molecular and functional characterisation of neuropeptides, obtaining novel insights into the evolution and comparative physiology of several neuropeptide signalling systems (*Cai et al., 2018*; *Elphick et al., 2018*; *Lin et al., 2017a*; *Lin et al., 2018*; *Odekunle et al., 2019*; *Semmens and Elphick, 2017*; *Tian et al., 2017*; *Tian et al., 2016*; *Tinoco et al., 2018*; *Yañez-Guerra et al., 2018*; *Yañez-Guerra et al., 2020*; *Zhang et al., 2020*). Accordingly, here we used *A. rubens* to enable the first detailed molecular, anatomical, and pharmacological analysis of SK/CCK-type signalling in an echinoderm.

## Results

### Cloning and sequencing of a cDNA encoding ArSK/CCKP

Analysis of *A. rubens* neural transcriptome sequence data has revealed the presence of an SK/CCK-type precursor in *A. rubens*, which was originally named ArCCKP (*Semmens et al., 2016*). Henceforth we refer to this precursor protein as ArSK/CCKP to reflect the sequence similarity that its constituent neuropeptides share with protostome SK-type peptides and chordate CCK/gastrin-type peptides. Here, cloning and sequencing of a cDNA encoding ArSK/CCKP confirmed the sequence obtained from transcriptome data (*Figure 1—figure supplement 1*).

### Identification of ArSK/CCKP-derived neuropeptides in extracts of *A. rubens* radial nerve cords

ArSK/CCKP comprises two putative SK/CCK-type neuropeptide sequences that are bounded by dibasic or tetrabasic cleavage sites. Both neuropeptide sequences have a C-terminal glycine residue, which is a potential substrate for post-translational amidation, and both neuropeptide sequences contain a tyrosine residue, which could be either sulphated or non-sulphated (ns) in the mature neuropeptides. Furthermore, an N-terminal glutamine residue (Q) in one of the neuropeptide sequences is a potential substrate for N-terminal pyroglutamylation (pQ) (*Figure 1—figure supplement 1*; *Figure 1—figure supplement 2a*).

LC-MS-MS analysis of *A. rubens* radial nerve cord extracts revealed the presence of four SK/CCK-type peptides derived from ArSK/CCKP: pQSKVDDY(SO$_3$H)GHGLFW-NH$_2$ (ArSK/CCK1; *Figure 1—figure supplement 2b*), pQSKVDDYGHGLFW-NH$_2$ (ArSK/CCK1(ns); *Figure 1—figure supplement 2c*), GGDDQY(SO$_3$H)GFGLFF-NH$_2$ (ArSK/CCK2; *Figure 1—figure supplement 2d*), and GGDDQYG-FGLFF-NH$_2$ (ArSK/CCK2(ns); *Figure 1—figure supplement 2e*). Thus, mass spectrometry confirmed that (i) the peptides are C-terminally amidated, (ii) the peptides are detected with or without tyrosine sulphation, and (iii) an N-terminal glutamine is post-translationally converted to pyroglutamate in the mature ArSK/CCK1 and ArSK/CCK1(ns) peptides.

Having determined the structures of SK/CCK-type neuropeptides derived from ArSK/CCKP, the sequences of ArSK/CCK1 and ArSK/CCK2 were aligned with the sequences of SK/CCK-type peptides that have been identified in other taxa (*Figure 1*). This revealed several evolutionarily conserved features, including a tyrosine residue (typically sulphated) and a C-terminal amide group that are separated by five to seven intervening residues. The C-terminal residue in most SK/CCK-type peptides, including ArSK/CCK2, is a phenylalanine residue. However, ArSK/CCK1 is atypical in having a C-terminal tryptophan residue, which is also a feature of an SK/CCK-type peptide in the bivalve mollusc *C. gigas*.

### Identification of an SK/CCK-type receptor in *A. rubens*

BLAST analysis of *A. rubens* neural transcriptome sequence data identified a transcript that encodes a 434-residue protein (ArSK/CCKR) that shares high sequence similarity with CCK-type receptors from other taxa (*Figure 2—figure supplement 1*). Phylogenetic analysis revealed that ArSK/CCKR groups within a clade including SK/CCK-type receptors that have been pharmacologically characterised in other taxa, including the human and mouse CCK/gastrin receptors CCKR1 and CCKR2, the *C. intestinalis* cionin receptors CioR1 and CioR2, the *D. melanogaster* sulfakinin receptors SKR1 and SKR2, and

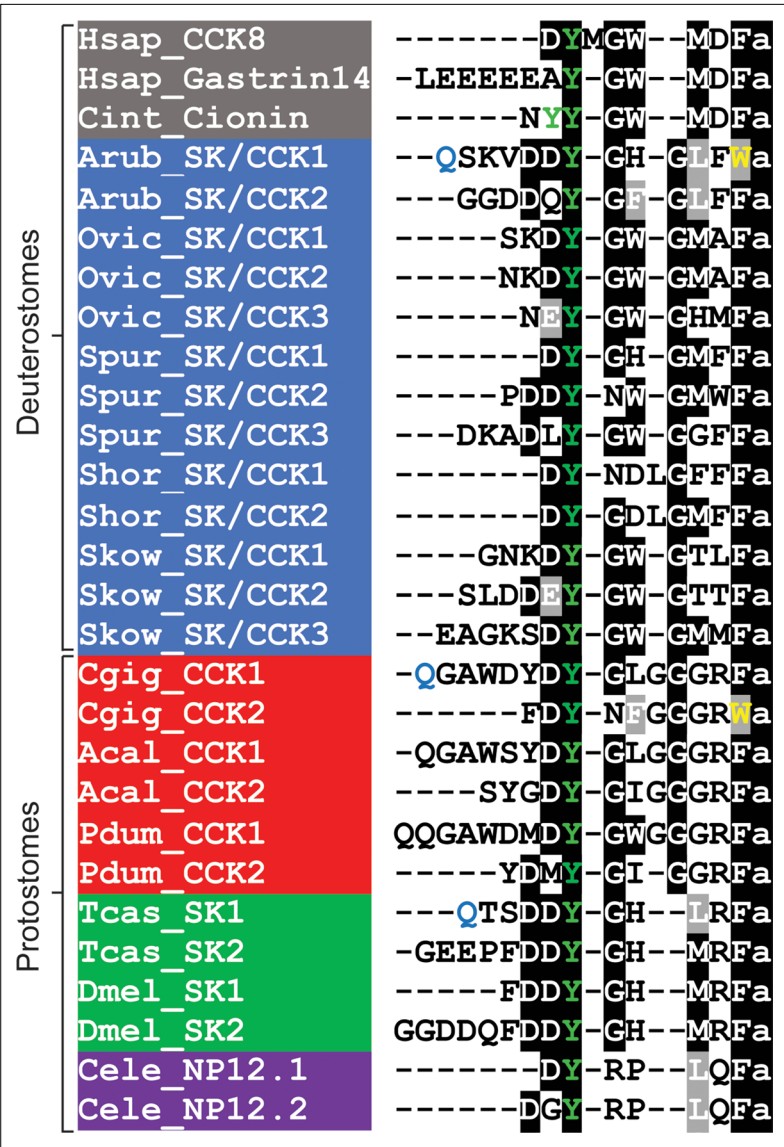

**Figure 1.** Comparison of the *Asterias rubens* sulfakinin/cholecystokinin (SK/CCK)-type neuropeptides ArSK/CCK1 and ArSK/CCK2 with SK/CCK-type neuropeptides from other taxa. Conserved residues are highlighted, with conservation in more than 70 % of sequences highlighted in black and conservative substitutions highlighted in grey. Experimentally verified conversion of an N-terminal glutamine residue (Q) to pyroglutamate in the mature peptide is indicated by the letter Q being shown in light blue. Tyrosine (Y) residues that are known or predicted to be subject to post-translational sulphation are shown in green. The C-terminal tryptophan (W) in ArSK/CCK1 and in a *Crassostrea gigas* SK/CCK-type peptide are shown in yellow to highlight that this feature is atypical of SK/CCK-type peptides. Predicted or experimentally verified C-terminal amides are shown as the letter 'a' in lowercase. Species names are highlighted in taxon-specific colours: grey (Chordata), blue (Ambulacraria), red (Lophotrochozoa), green (Arthropoda),and purple (Nematoda). Abbreviations are as follows: Acal (*Aplysia californica*), Arub (*Asterias rubens*), Cele (*Caenorhabditis elegans*), Cgig (*Crassostrea gigas*), Cint (*Ciona intestinalis*), Dmel (*Drosophila melanogaster*), Hsap (*Homo sapiens*), Ovic (*Ophionotus victoriae*), Pdum (*Platynereis dumerilii*), Shor (*Stichopus horrens*), Skow (*Saccoglossus kowalevskii*), Spur (*Strongylocentrotus purpuratus*), Tcas (*Tribolium castaneum*). The accession numbers of the sequences are listed in *Figure 1—source data 1*. The nucleotide sequence of a cloned cDNA encoding the *A. rubens* SK/CCK-type precursor is shown in *Figure 1— figure supplement 1*. Mass spectroscopic analysis of the structures of the peptides derived from the *A. rubens* SK/CCK-type precursor is presented in *Figure 1—figure supplement 2*. The raw data for the results shown in *Figure 1—figure supplement 2* can be found in *Figure 1—figure supplement 2—source data 1*.

The online version of this article includes the following source data and figure supplement(s) for figure 1:

*Figure 1 continued*

**Source data 1.** Accession numbers for precursors of the neuropeptides shown in the sequence alignment in *Figure 1*.

**Figure supplement 1.** The *Asterias rubens* sulfakinin/cholecystokinin (SK/CCK)-type precursor (ArSK/CCKP).

**Figure supplement 2.** Determination of the structures of peptides derived from ArSK/CCKP by mass spectrometric (LC-MS-MS) analysis of *Asterias rubens* radial nerve cord extract.

**Figure supplement 2—source data 1.** Data for the mass spectra shown in *Figure 1—figure supplement 2*.

the recently characterised *C. gigas* receptors CCKR1 and CCKR2 (*Figure 2*). Thus, this demonstrates that ArSK/CCKR is an ortholog of SK/CCK-type receptors that have been characterised in other taxa. Furthermore, reflecting known animal phylogenetic relationships, ArSK/CCKR is positioned within a branch of the tree that comprises SK/CCK-type receptors from deuterostomes, and more specifically it is positioned within an ambulacrarian clade that comprises SK/CCK-type receptors from other echinoderms and from the hemichordate *Saccoglossus kowalevskii*.

Analysis of the amino acid sequence of ArSK/CCKR using the Protter tool (*Omasits et al., 2014*) revealed seven predicted transmembrane domains, as expected for a GPCR, and three potential N-glycosylation sites in the predicted extracellular N-terminal region of the receptor (*Figure 2— figure supplement 2*).

## ArSK/CCK1 and ArSK/CCK2 are ligands for ArSK/CCKR

Previous studies on other species have revealed that sulphation of the tyrosine residue in CCK-type peptides is often important for receptor activation and bioactivity (*Dufresne et al., 2006*; *Kubiak et al., 2002*; *Schwartz et al., 2018*; *Sekiguchi et al., 2012*; *Yu et al., 2015*). Accordingly, neuropeptides derived from ArSK/CCKP were detected in *A. rubens* radial nerve cord extracts with sulphated tyrosines (ArSK/CCK1 and ArSK/CCK2; *Figure 1—figure supplement 2b, d*). Therefore, the sulphated peptides ArSK/CCK1 and ArSK/CCK2 were synthesized and tested as ligands for ArSK/CCKR. However, because non-sulphated forms of ArSK/CCK1 (ArSK/CCK1(ns)) and ArSK/CCK2 (ArSK/CCK(ns)) were also detected in *A. rubens* radial nerve extracts (*Figure 1—figure supplement 2c, e*), we also synthesized and tested ArSK/CCK2(ns) to investigate if absence of tyrosine sulphation affects receptor activation. Using Chinese hamster ovary (CHO)-K1 cells expressing aequorin as an assay system, ArSK/CCK1, ArSK/CCK2, and ArSK/CCK2(ns) did not elicit luminescence responses when tested on cells transfected with an empty vector (*Figure 3*). However, all three peptides caused concentration-dependent stimulation of luminescence in CHO-K1 cells transfected with ArSK/CCKR (*Figure 3*). The $EC_{50}$ values for ArSK/CCK1, ArSK/CCK2, and ArSK/CCK2(ns) were 0.25 nM (*Figure 3a*), 0.12 nM (*Figure 3b*), and 48 µM (*Figure 3c*), respectively. Thus, although activation of ArSK/CCKR was observed in vitro with ArSK/CCK2(ns) (*Figure 3c*), this peptide is five to six orders of magnitude less potent than ArSK/CCK1 and ArSK/CCK2 as a ligand for ArSK/CCKR. This indicates that the non-sulphated peptides ArSK/CCK1(ns) and ArSK/CCK2(ns) that were detected in *A. rubens* radial nerve extracts are unlikely to have physiological effects in vivo. Furthermore, two other *A. rubens* neuropeptides that share modest C-terminal sequence similarity with the *A. rubens* SK/CCK-type peptides – the SALMFamide neuropeptide S2 (SGPYSFNSGLTF-NH₂) and the tachykinin-like peptide ArTK2 (GGGVPHVFQSGGIF-NH₂; also referred to as ArGxFamide2) – were found to be inactive when tested as ligands for ArSK/CCKR at concentrations ranging from $10^{-12}$ to $10^{-4}$ M (*Figure 3—figure supplement 1*), demonstrating the specificity of ArSK/CCK1 and ArSK/CCK2 as ligands for ArSK/CCKR.

## Localisation of ArSK/CCKP expression in *A. rubens* using mRNA in situ hybridization

To gain anatomical insights into the physiological roles of SK/CCK-type neuropeptides in starfish, mRNA in situ hybridisation methods were employed to enable analysis of the distribution of the ArSK/CCKP transcript in *A. rubens*. As described below and illustrated in *Figure 4*, expression of ArSK/CCKP was observed in the central nervous system (CNS), digestive system, body wall, and tube feet.

The CNS of *A. rubens* comprises radial nerve cords that extend along the oral side of each arm, with two rows of tube feet (locomotory organs) on either side. The five radial nerve cords are linked by a circumoral nerve ring in the central disk (*Pentreath and Cobb, 1972*). Analysis of ArSK/CCKP mRNA

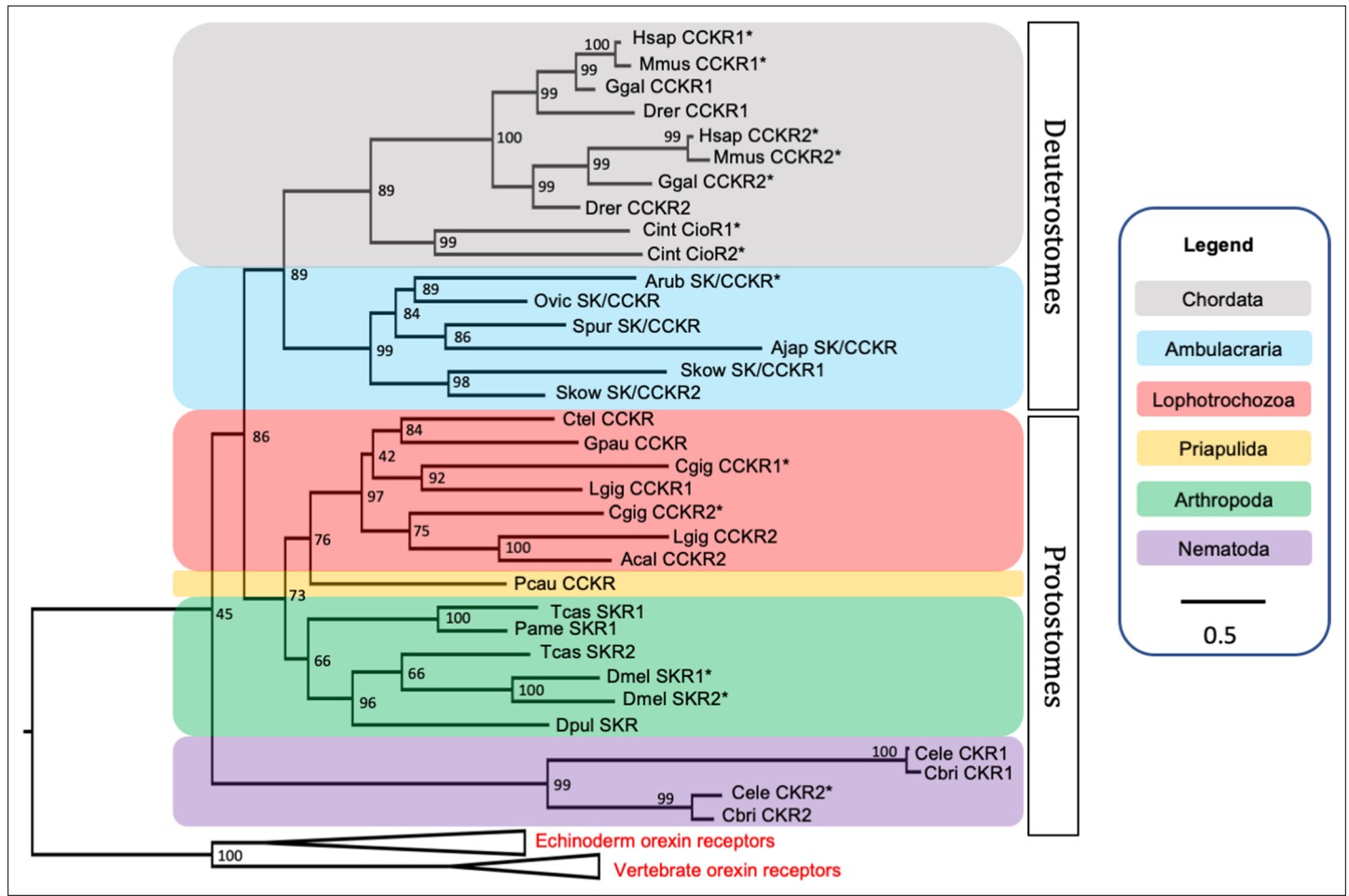

**Figure 2.** Phylogenetic tree showing that the predicted *Asterias rubens* sulfakinin/cholecystokinin (SK/CCK)-type receptor (ArSK/CCKR) is an ortholog of SK/CCK-type receptors in other taxa, which include receptors (*) for which the peptide ligands have been identified experimentally. The tree was generated using the maximum likelihood method (*Guindon et al., 2009*), with the percentage of replicate trees in which two or more sequences form a clade in a bootstrap test (1000 replicates) shown at the node of each clade (*Zhaxybayeva and Gogarten, 2002*). Orexin receptors were included as an outgroup. The tree is drawn to scale, with branch lengths in the same units as those of the evolutionary distances used to infer the phylogenetic tree. Evolutionary analyses were conducted using the IQ-tree server (*Trifinopoulos et al., 2016*). Taxa are colour-coded as explained in the key. Abbreviations of species names are as follows: Acal (*Aplysia californica*), Ajap (*Apostichopus japonicus*), Arub (*Asterias rubens*), Cbri (*Caenorhabditis briggsae*), Cele (*Caenorhabditis elegans*), Cgig (*Crassostrea gigas*), Cint (*Ciona intestinalis*), Ctel (*Capitella teleta*), Dmel (*Drosophila melanogaster*), Dpul (*Daphnia pulex*), Drer (*Danio rerio*), Ggal (*Gallus gallus*), Gpau (*Glossoscolex paulistus*), Hsap (*Homo sapiens*), Lgig (*Lottia gigantea*), Mmus (*Mus musculus*), Ovic (*Ophionotus victoriae*), Pame (*Periplaneta americana*), Pcau (*Priapulus caudatus*), Skow (*Saccoglossus kowalevskii*), Spur (*Strongylocentrotus purpuratus*), Tcas (*Tribolium castaneum*). The accession numbers of the sequences used for this phylogenetic tree are listed in *Figure 2—source data 1*. The nucleotide sequence and the predicted topology of ArSK/CCKR are shown in *Figure 2—figure supplement 1* and *Figure 2—figure supplement 2*, respectively.

The online version of this article includes the following figure supplement(s) for figure 2:

**Source data 1.** Accession numbers for the receptor sequences used for the phylogenetic tree in *Figure 2*.

**Figure supplement 1.** The *Asterias rubens* sulfakinin/cholecystokinin (SK/CCK)-type receptor (ArSK/CCKR).

**Figure supplement 2.** Topology of ArSK/CCKR.

expression revealed stained cells in both the ectoneural and hyponeural regions of the radial nerve cords (*Figure 4a*) and circumoral nerve ring (*Figure 4b and c*). Furthermore, the specificity of staining observed with antisense probes was confirmed by an absence of staining in tests with sense probes (*Figure 4a*). Stained cells were also revealed in the ectoneural segmental branches of the radial nerve cords (*Figure 4d*) and in the marginal nerves (*Figure 4e*), which run parallel with the radial nerve cords lateral to the outer row of tube feet. ArSK/CCKP-expressing cells were also revealed in tube feet, with stained cells located in the podium proximal to its junction with the radial and marginal nerves

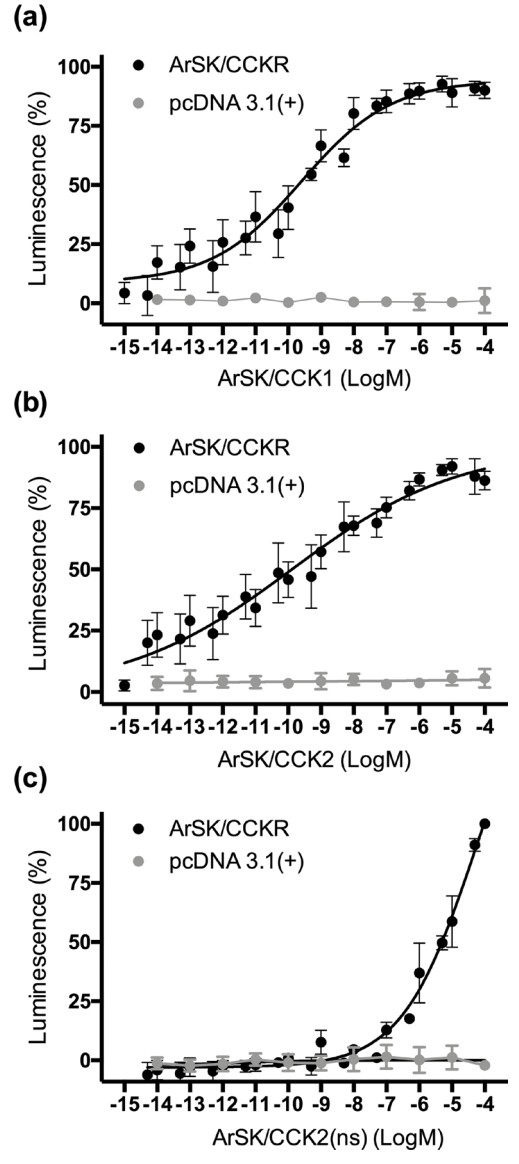

**Figure 3.** Experimental demonstration that the *Asterias rubens* SK/CCK-type peptides ArSK/CCK1 and ArSK/CCK2 act as ligands for the *A. rubens* SK/CCK-type receptor ArSK/CCKR. The sulphated peptides ArSK/CCK1 (**a**), ArSK/CCK2 (**b**), and the non-sulphated peptide ArSK/CCK2(ns) (**c**) trigger dose-dependent luminescence in Chinese hamster ovary (CHO)-K1 cells stably expressing mitochondrial targeted apoaequorin (G5A) that were co-transfected with plasmids encoding the promiscuous human G-protein Gα16 and ArSK/CCKR (black). Control experiments where cells were transfected with an empty pcDNA 3.1(+) vector are shown in grey. Each point represents mean values (± s.e.m.) from at least four independent experiments performed in triplicate. Luminescence is expressed as a percentage of the maximal response observed in each experiment. The EC₅₀ values for ArSK/CCK1 (**a**) and ArSK/CCK2 (**b**) are 0.25 and 0.12 nM, respectively.

*Figure 3 continued on next page*

*Figure 3 continued*

In comparison, the absence of tyrosine (Y) sulphation in ArSK/CCK2(ns) (**c**) causes a massive loss of potency ($EC_{50}$ = 48 µM), indicating that the sulphated peptides act as ligands for ArSK/CCKR physiologically. A graph showing the selectivity of ArSK/CCKR as a receptor for ArSK/CCK-type peptides is presented in *Figure 3—figure supplement 1*. See *Figure 3—source data 1* for source data.

The online version of this article includes the following figure supplement(s) for figure 3:

**Source data 1.** Data for the graphs shown in *Figure 3* and *Figure 3—figure supplement 1*.

**Figure supplement 1.** Graph showing the selectivity of ArSK/CCKR as a receptor for sulfakinin/cholecystokinin (SK/CCK)-type peptides.

---

(*Figure 4e*) and in the tube foot disk (*Figure 4f*). In the digestive system, ArSK/CCKP-expressing cells were revealed in the mucosa of the oesophagus (*Figure 4g*), cardiac stomach (*Figure 4i and h*), pyloric stomach (*Figure 4i*), pyloric ducts (*Figure 4j*), pyloric caeca (*Figure 4j*), and intestine (*Figure 4k and l*). ArSK/CCKP expressing cells were also revealed in the external epithelium of the body wall (*Figure 4m and n*).

## Immunohistochemical localisation of ArSK/CCK1 in *A. rubens*

Use of mRNA in situ hybridisation (see above) revealed the location of cells expressing ArSK/CCKP in *A. rubens*. However, a limitation of this technique is that it does not reveal the axonal processes of peptidergic neurons. Therefore, to enable this using immunohistochemistry, we generated and affinity-purified rabbit antibodies to ArSK/CCK1. Enzyme-linked immunosorbent assay (ELISA) analysis of antiserum revealed the presence of antibodies to the ArSK/CCK1 peptide antigen (*Figure 5—figure supplement 1a*) and ELISA analysis of affinity-purified antibodies to the ArSK/CCK1 antigen peptide revealed the specificity of these antibodies for ArSK/CCK1 because they do not cross-react with ArSK/CCK2, ArSK/CCK2(ns), or the starfish luqin-type neuropeptide ArLQ (*Figure 5—figure supplement 1b*). Immunohistochemical tests with affinity-purified ArSK/CCK1 antibodies revealed extensive immunostaining in sections of *A. rubens*, as described in detail below and illustrated in *Figure 5*.

ArSK/CCK1-immunoreactive (ArSK/CCK1-ir) cells were revealed in the ectoneural and hyponeural regions of the radial nerve cords (*Figure 5a and b*). Furthermore, dense networks of immunostained fibres were revealed in the ectoneural

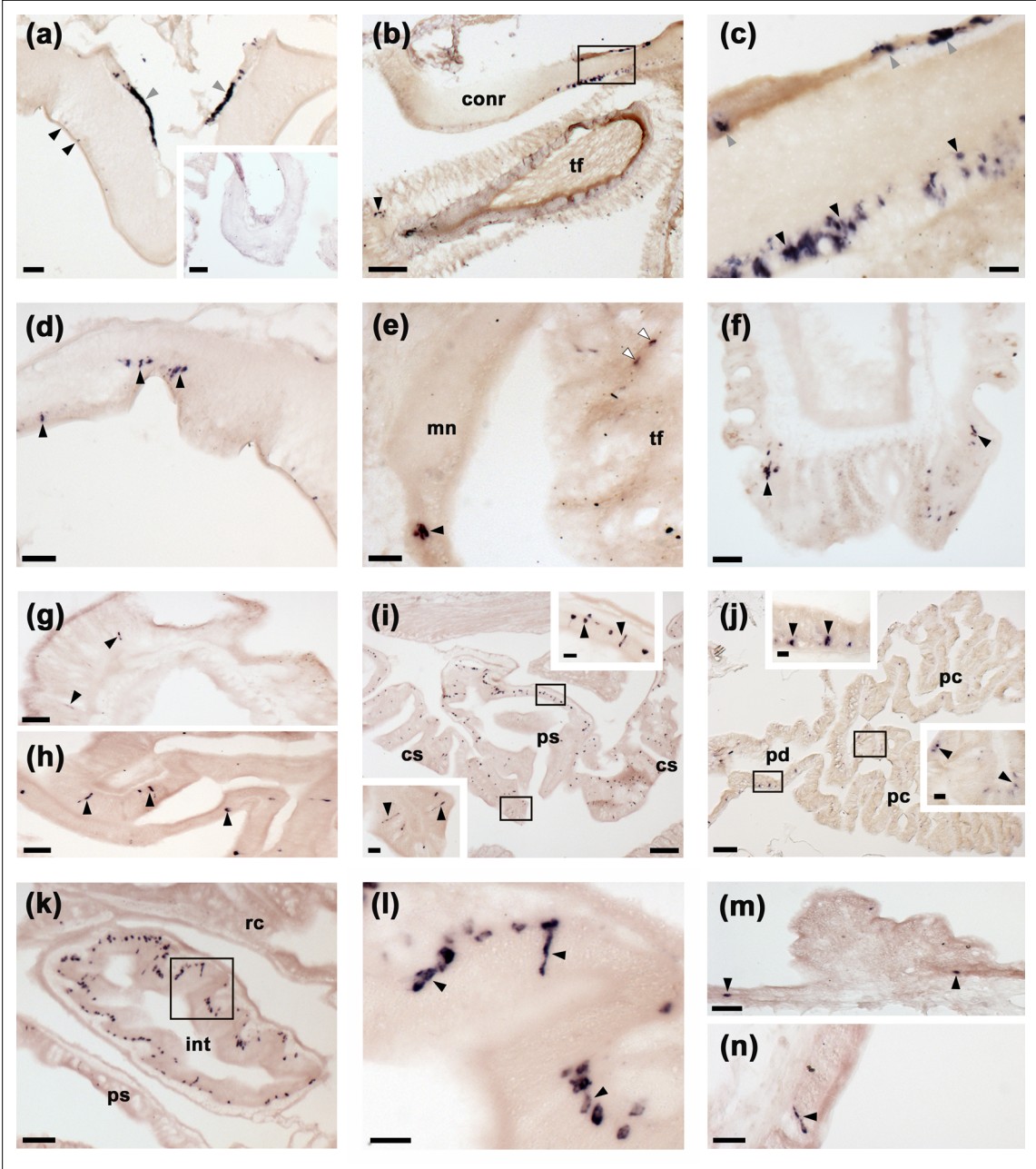

**Figure 4.** Localisation of ArSK/CCKP expression in *Asterias rubens* using mRNA in situ hybridization. (**a**) Using antisense probes, ArSK/CCKP-expressing cells are revealed in the ectoneural (black arrowheads) and hyponeural (grey arrowheads) regions of a radial nerve cord. The specificity of staining with antisense probes is demonstrated by the absence of staining in a radial nerve cord section incubated with sense probes (see inset). (**b**) ArSK/CCKP-expressing cells in the circumoral nerve ring (see boxed area) and in the disk region of a peri-oral tube foot (black arrowhead). (**c**) High magnification image of the boxed area in (**b**), showing stained cells in the ectoneural (black arrowheads) and hyponeural (grey arrowheads) regions of the circumoral nerve ring. (**d**) ArSK/CCKP-expressing cells in a lateral branch of the radial nerve cord (black arrowheads). (**e**) ArSK/CCKP-expressing cells adjacent to the marginal nerve (black arrowhead) and in the stem of a tube foot (white arrowheads). (**f**) ArSK/CCKP-expressing cells (black arrowheads) adjacent to the basal nerve ring in the disk region of a tube foot. (**g**) ArSK/CCKP-expressing cells (black arrowheads) in the mucosal layer of the oesophagus. (**h**) ArSK/CCKP-expressing cells (black arrowheads) in the mucosal layer of the cardiac stomach. (**i**) ArSK/CCKP-expressing cells (black arrowheads) in the cardiac stomach and pyloric stomach, with the boxed regions shown at higher magnification in the insets. (**j**) ArSK/CCKP-expressing cells (black arrowheads) in the pyloric duct and pyloric caeca, with the boxed regions shown at higher magnification in the insets. (**k,l**) ArSK/CCKP-expressing cells (black arrowheads) in an oblique section of the intestine; the boxed region in (**k**) is shown at higher magnification in (**l**). (**m,n**) ArSK/CCKP-expressing cells (black arrowheads) in the external epithelium of the body wall. Abbreviations: conr, circumoral nerve ring; int, intestine; mn, marginal nerve; pc, pyloric caecum; pd, pyloric duct; ps, pyloric stomach; rc, rectal caeca; tf, tube foot. Scale bars: (**b**), (**i**), (**j**) = 120 µm; (**a**), (a-inset), (**k**) = 60 µm; (**d**), (**e**), (**f**), (**g**), (**h**), (**m**) = 32 µm; (**c**), (i-insets), (j-insets), (**l**), (**n**) = 16 µm.

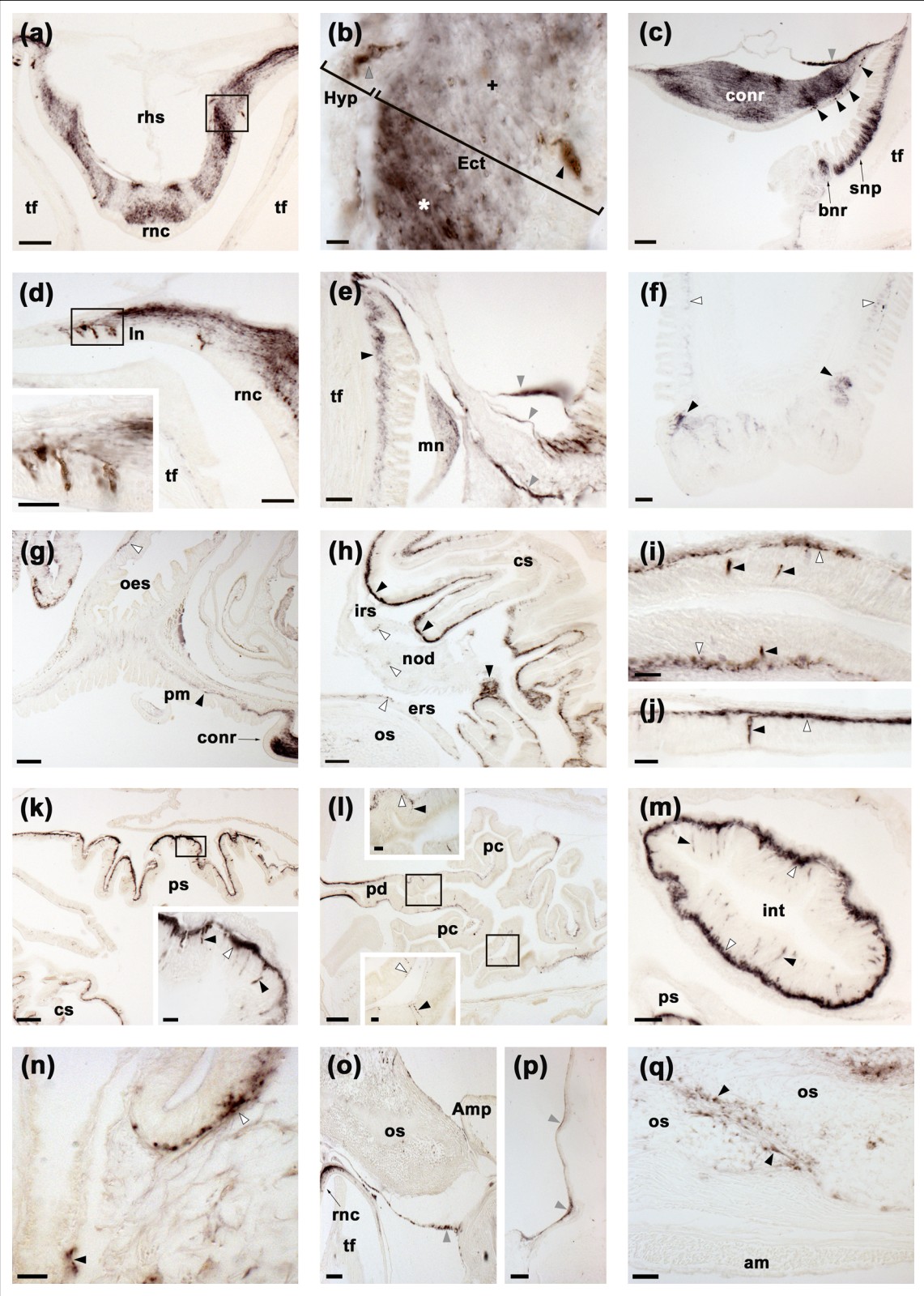

**Figure 5.** Localisation of ArSK/CCK1 expression in *Asterias rubens* using immunohistochemistry. (**a**) ArSK/CCK1-immunoreactivity (ArSK/CCK1-ir) in a transverse section of the V-shaped radial nerve cord, with bilaterally symmetrical regional variation in the density of immunostaining in the ectoneural neuropile. (**b**) High magnification image of the boxed region in (**a**), showing stained cell bodies in the hyponeural (grey arrowhead) and ectoneural (black arrowhead) regions of the radial nerve cord. Regions of the ectoneural neuropile containing a higher (*) and lower (+) densities of immunostained

*Figure 5 continued on next page*

*Figure 5 continued*

fibres can be seen here. (**c**) ArSK/CCK1-ir in the circumoral nerve ring, with stained cells present in the hyponeural region (grey arrowheads) and in the ectoneural epithelium (black arrowheads); in the ectoneural neuropile there is regional variation in the density of immunostained fibres. Immunostaining can also be seen here in the sub-epithelial nerve plexus and basal nerve ring of an adjacent peri-oral tube foot. (**d**) ArSK/CCK1-ir in cells and fibres in a lateral branch of the radial nerve cord; the inset shows immunostained cells in the boxed region at higher magnification. (**e**) ArSK/CCK1-ir in the marginal nerve, the sub-epithelial nerve plexus of an adjacent tube foot (black arrowhead) and in branches of the lateral motor nerve (grey arrowheads). (**f**) ArSK/CCK1-ir in the sub-epithelial nerve plexus (white arrowheads) and basal nerve ring (black arrowheads) of a tube foot. (**g**) ArSK/CCK1-ir in the basi-epithelial nerve plexus of the peristomial membrane (black arrowhead) and the oesophagus (white arrowhead); immunostaining in the circumoral nerve can also be seen here. (**h**) ArSK/CCK1-ir in the lateral pouches of the cardiac stomach; note that the density of immunostained fibres is highest (black arrowheads) in regions of the mucosa adjacent to the intrinsic retractor strand; immunostaining in the intrinsic retractor strand, nodule and extrinsic retractor strand can also be seen here (white arrowheads). (**i,j**) High magnification images of cardiac stomach tissue showing ArSK/CCK1-ir in cell bodies (black arrowheads) and their processes in the basi-epithelial nerve plexus (white arrowheads); note that in (**j**) a process emanating from an immunostained cell body can be seen projecting into the plexus. (**k**) ArSK/CCK1-ir in the cardiac stomach and pyloric stomach; the boxed region is shown at higher magnification in the inset, showing immunostaining in cells (black arrowheads) and the basi-epithelial nerve plexus (white arrowhead). (**l**) ArSK/CCK1-ir in the pyloric duct and pyloric caeca; the boxed regions are shown at higher magnification in the insets, where immunostained cells (black arrowheads) and fibres (white arrowheads) can be seen. (**m**) ArSK/CCK1-ir in an oblique section of the intestine, with immunostained cells in the mucosa (black arrowheads) and intense immunostaining in the basi-epithelial nerve plexus (white arrowheads). (**n**) ArSK/CCK1-ir in the basi-epithelial nerve plexus of the body wall external epithelium (white arrowhead) and in the lining of a papula (black arrowhead). (**o**) ArSK/CCK1-ir in nerve fibres projecting around the base of a tube foot at its junction with the neck of its ampulla. (**p**) ArSK/CCK1-ir in nerve fibres located in the coelomic lining of the lateral region of the body wall. (**q**) ArSK/CCK1-ir in inter-ossicular tissue of the body wall. Abbreviations: am, apical muscle; Amp, ampulla; bnr, basal nerve ring; conr, circumoral nerve ring; cs, cardiac stomach; Ect, ectoneural; ers, extrinsic retractor strand; Hyp, hyponeural; int, intestine; irs, intrinsic retractor strand; ln, lateral nerve; mn, marginal nerve; nod, nodule; oes, oesophagus; os, ossicle; pm, peristomial membrane; pc, pyloric caecum; pd, pyloric duct; ps, pyloric stomach; rhs, radial hemal strand; rnc, radial nerve cord; snp, sub-epithelial nerve plexus; tf, tube foot. Scale bars: (**g**), (**k**), (**l**) = 120 μm; (**a**), (**c**), (**f**), (**h**), (**o**), (**p**) = 60 μm; (**d**), (**e**), (**m**), (**q**) = 32 μm; (d-inset), (**i**), (**j**), (k-inset), (l-insets), (**n**) = 16 μm; (**b**) = 6 μm. Graphs showing enzyme-linked immunosorbent assay (ELISA)-based characterisation of the antibodies to ArSK/CCK1 used here for immunohistochemistry are presented in *Figure 5—figure supplement 1*.

The online version of this article includes the following figure supplement(s) for figure 5:

**Figure supplement 1.** Characterisation of a rabbit antibodies to ArSK/CCK1 using an enzyme-linked immunosorbent assay (ELISA).

---

neuropile, with bilaterally symmetrical regional variation in the density of immunostaining (*Figure 5b*). Likewise, ArSK/CCK1-ir cells were revealed in the ectoneural and hyponeural regions of the circumoral nerve ring, also with regional variation in the density of immunostained fibres in the ectoneural neuropile (*Figure 5c*). Immunostained cells and/or processes were also revealed in the segmental lateral branches of the radial nerve cords (*Figure 5d*) and in the marginal nerve cords (*Figure 5e*). Consistent with the expression of ArSK/CCKP/ArSK/CCK1 in the hyponeural region of radial nerve cords, ArSK/CCK1-ir fibres were revealed in the lateral motor nerves (*Figure 5e*). In tube feet, ArSK/CCK1-ir fibres were revealed in the sub-epithelial nerve plexus of the podium and in the basal nerve ring of the disk region (*Figure 5c and f*).

ArSK/CCK1-ir cells and/or fibres were revealed in the mucosa and basiepithelial nerve plexus, respectively, of many regions of the digestive system, including the peristomial membrane (*Figure 5g*), oesophagus (*Figure 5g*), cardiac stomach (*Figure 5h, i and j*), pyloric stomach (*Figure 5k*), pyloric ducts (*Figure 5l*), pyloric caeca (*Figure 5l*), and intestine (*Figure 5m*). Consistent with patterns of ArSK/CCKP transcript expression (see above), regional differences in the abundance of stained cells and fibres were observed. Regions of the digestive system containing denser populations of ArSK/CCK1-ir cells and/or fibres include the lateral pouches of the cardiac stomach (*Figure 5h*), the roof of the pyloric stomach (*Figure 5k*), and the intestine (*Figure 5m*).

Immunostaining was observed in the sub-epithelial nerve plexus of the external epithelium of the body wall (*Figure 5n*) and in papulae (*Figure 5n*), which are protractable appendages that penetrate through the body wall to enable gas exchange between external seawater and the coelomic fluid (*Cobb, 1978*). Immunostained fibres were also observed in branches of the lateral motor nerves located in the coelomic lining of the body wall (*Figure 5o and p*). However, no staining was observed in the apical muscle – a thickening of longitudinally orientated muscle that is located under the coelomic epithelial layer of the body wall along the aboral midline of each arm (*Figure 5q*). The bulk of the body wall in *A. rubens* is comprised of calcite ossicles that are interconnected by muscles and collagenous tissue and ArSK/CCK1-ir fibres were revealed in the inter-ossicular tissue (*Figure 5q*).

## Comparison of ArSK/CCK1 and ArSK/CCKR expression in *A. rubens* using double immunofluorescence labelling

Having mapped expression ArSK/CCK1 in *A. rubens* using rabbit primary antibodies, we then generated antibodies to ArSK/CCKR in a guinea pig so that double labelling methods could be employed to directly compare the distribution of ArSK/CCK1 and its cognate receptor ArSK/CCKR in *A. rubens*. ELISA analysis of the guinea pig antiserum revealed the presence of antibodies to an antigen peptide corresponding to a 15 amino acid sequence at the C-terminus of ArSK/CCKR (*Figure 6—figure supplement 1*). Antibodies to this peptide antigen were then affinity-purified and used together with affinity-purified rabbit antibodies to ArSK/CCK1 for double immunofluorescence labelling. As described below and illustrated in *Figure 6*, this enabled comparison of the distribution of ArSK/CCK1 (green) and ArSK/CCK-R (red) in the CNS, tube feet, apical muscle, and digestive system of *A. rubens*.

In the CNS, ArSK/CCK1-ir and ArSK/CCKR-ir were revealed in both the ectoneural and hyponeural regions of the circumoral nerve ring (*Figure 6a*) and radial nerve cords (*Figure 6b and c*). Complementary patterns of immunostaining were observed, with ArSK/CCK1-ir and ArSK/CCKR-ir generally detected in different populations of cells and fibres. Thus, distinct clusters of ArSK/CCK1-ir and ArSK/CCKR-ir cell bodies can be observed in the hyponeural region of the radial nerve cord (*Figure 6b*). Furthermore, in the ectoneural neuropile a 'salt and pepper' pattern of immunolabelling can be observed, with ArSK/CCK1-ir fibres (green) interspersed with ArSK/CCKR-ir fibres (red) in the circumoral nerve ring (*Figure 6a*) and radial nerve cords (*Figure 6b and c*). However, the presence of yellow/orange labelling in some cells/fibres indicates that ArSK/CCKR may be co-expressed in a sub-population ArSK/CCK1-expressing neurons (*Figure 6a–c*).

In tube feet, ArSK/CCK1-ir and ArSK/CCKR-ir processes were revealed in the sub-epithelial nerve plexus of the podium (*Figure 6d*) and in the basal nerve ring of the disk region (data not shown). In the apical muscle, ArSK/CCK1-ir processes and ArSK/CCKR-ir cells/processes were revealed predominantly in the coelomic epithelial layer and sub-epithelial nerve plexus, but ArSK/CCK1-ir and ArSK/CCKR-ir processes were also found to be sparsely distributed in the underlying muscle fibre containing layer (*Figure 6e*).

In the cardiac stomach (*Figure 6f*) and pyloric stomach (*Figure 6h*), ArSK/CCK1-ir and ArSK/CCKR-ir cells and/or fibres were revealed in the mucosa and basiepithelial nerve plexus, and prominent ArSK/CCKR-ir could also be observed in the visceral muscle layer. As in the CNS, tube feet, and apical muscle, analysis of immunostaining in the basiepithelial nerve plexus of the cardiac and pyloric stomach indicates that ArSK/CCK1 and ArSK/CCKR are largely localised in different fibre populations but often in close proximity to each other (*Figure 6f and h*). However, in the nodule region that links the cardiac stomach to extrinsic retractor strands, evidence of co-expression of ArSK/CCK1 and ArSK/CCKR was also obtained (*Figure 6f*).

## ArSK/CCK1 and ArSK/CCK2 cause concentration-dependent contraction of in vitro cardiac stomach, tube foot, and apical muscle preparations from *A. rubens*

Informed by the localisation of ArSK/CCKP/ArSK/CCK1 and ArSK/CCK-R expression in the cardiac stomach, tube feet, and apical muscle of *A. rubens*, we tested the effects of ArSK/CCKP-derived neuropeptides on in vitro preparations of these organs. Both ArSK/CCK1 and ArSK/CCK2 caused concentration-dependent contraction of cardiac stomach preparations when tested at concentrations ranging from 1 nM to 1 µM (*Figure 7a and b*). ArSK/CCK2(ns) caused modest contraction of cardiac stomach preparations by comparison with ArSK/CCK2 (*Figure 7b*), and comparison of the ArSK/CCK2 and ArSK/CCK2(ns) data using a two-way ANOVA revealed a significant difference in the effects of the peptides on cardiac stomach preparations, irrespective of concentration ($p < 0.0001$). However, two-way ANOVA revealed no significant difference in the effects of ArSK/CCK1 and ArSK/CCK2 on cardiac stomach preparations. To enable normalisation of the effects of the ArSK/CCKP-derived peptides between experiments, the neuropeptide NGFFYamide was also tested on each preparation at a concentration of 100 nM (*Figure 7a*), and at this concentration the effects of ArSK/CCK1 and ArSK/CCK2 (1 µM) were not significantly different (Student's t-test; $p > 0.05$) to the effect of NGFFYamide (data not shown).

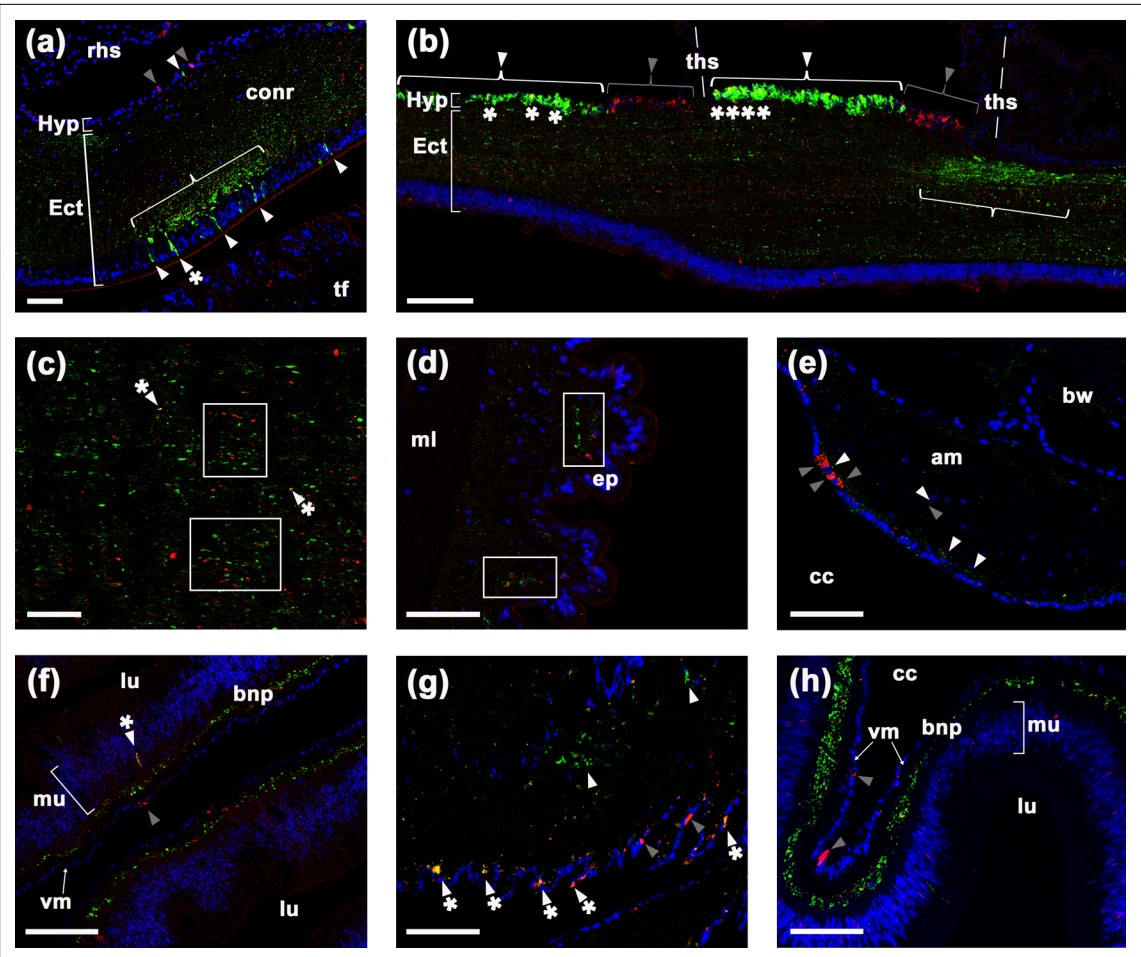

**Figure 6.** Comparison of ArSK/CCK1 and ArSK/CCK receptor (ArSK/CCKR) expression in *Asterias rubens* using double immunofluorescence labelling. (**a**) ArSK/CCK1-immunoreactivity (ArSK/CCK1-ir; green) and ArSK/CCKR immunoreactivity (ArSK/CCK-R-ir; red) in a transverse section of the circumoral nerve ring, with cell nuclei stained using DAPI (blue). Fibres exhibiting ArSK/CCK1-ir or ArSK/CCKR-ir can be seen throughout the neuropile of the ectoneural region. However, there is variation in the density of immunostained fibres/cells and a bracketed neuropile region containing a higher density of ArSK/CCK1-ir fibres is located proximal to a cluster of cells in the ectoneural epithelium that exhibit ArSK/CCK1-ir (green; white arrowheads) or ArSK/CCK1-ir and ArSK/CCKR-ir (green and yellow; white arrowheads with asterisk). Cells exhibiting ArSK/CCK1-ir (white arrowhead) or ArSK/CCKR-ir (grey arrowhead) can be seen in the hyponeural region. (**b**) Longitudinal section of a radial nerve cord showing segmentally repeated clusters of labelled cells in the hyponeural region, with each segment bounded by transverse hemal strands. Note the distinct clusters of cells exhibiting ArSK/CCKR-ir (grey arrowhead) and cells exhibiting ArSK/CCK1-ir (white arrowhead). However, the presence of yellow regions in some of the cells exhibiting ArSK/CCK1-ir (asterisks) indicates that ArSK/CCKR may be co-expressed in these cells. As in the circumoral nerve (**a**), there is variation in the density of immunostained fibres in the ectoneural neuropile and a region with a high density of fibres exhibiting ArSK/CCK1-ir is bracketed. (**c**) High magnification image of a longitudinal section of a radial nerve cord showing a 'salt and pepper' pattern of labelling in the ectoneural neuropile, consistent with expression of ArSK/CCK1 and ArSK/CCKR in different but often adjacent populations of fibres (see white rectangles). However, the presence of yellow labelling (white arrowhead with asterisk) may be indicative of co-expression in a few fibres. (**d**) Immunostained processes exhibiting ArSK/CCK1-ir or ArSK/CCK-R-ir can be seen here in close proximity (white rectangles) within the sub-epithelial nerve plexus of a tube foot podium. (**e**) Immunostained cells/processes exhibiting ArSK/CCK1-ir (white arrowheads) or ArSK/CCKR-ir (grey arrowheads) can be seen here in the coelomic epithelial layer, sub-epithelial nerve plexus, or muscle fibre layer of an apical muscle. (**f**) High magnification image of the lateral pouches of cardiac stomach showing fibres exhibiting ArSK/CCK1-ir or ArSK/CCKR-ir in the basiepithelial nerve plexus. Cells exhibiting ArSK/CCKR-ir can be seen in the visceral muscle layer (grey arrowhead) and a cell that exhibits both ArSK/CCK1-ir and ArSK/CCKR-ir can be seen in the mucosal layer (white arrowhead with asterisk). (**g**) Immunostained cells/processes exhibiting ArSK/CCK1-ir (white arrowhead) or ArSK/CCK-R-ir (grey arrowhead) or both ArSK/CCK1-ir and ArSK/CCK-R-ir (white arrowhead with asterisks) in a nodule that links the cardiac stomach to extrinsic retractor strands. (**h**) High magnification image of pyloric stomach showing fibres exhibiting ArSK/CCK1-ir or ArSK/CCKR-ir in the basiepithelial nerve plexus. Cells exhibiting ArSK/CCKR-ir can be seen in the visceral muscle layer (grey arrowhead). Abbreviations: am, apical muscle; bnp, basiepithelial nerve plexus; bw, body wall; cc, coelomic cavity; conr, circumoral nerve ring; Ect, ectoneural; ep, epithelium; Hyp, hyponeural; ml, muscle layer; mu, mucosa; rhs, radial hemal strand; tf, tube foot; ths, transverse hemal strand; vm, visceral muscle layer. Scale bars: (**b**) = 70 µm; (**a**), (**d**), (**e**), (**f**), (**g**), (**h**) = 40 µm; (**c**) = 20 µm.

The online version of this article includes the following figure supplement(s) for figure 6:

*Figure 6 continued on next page*

*Figure 6 continued*

**Figure supplement 1.** Characterisation of rabbit antibodies to ArSK/CCKR using an enzyme-linked immunosorbent assay (ELISA).

Consistent with the effects of ArSK/CCKP-derived neuropeptides on cardiac stomach preparations, ArSK/CCK1 and ArSK/CCK2 (1 nM to 1 µM) also caused concentration-dependent contraction of tube foot preparations (*Figure 7c and d*). Furthermore, comparison of the effects of ArSK/CCK1 and ArSK/CCK2 on tube feet using a two-way ANOVA revealed significant differences, irrespective of concentration (p < 0.0001). In addition, Bonferroni's multiple comparison test showed that ArSK/CCK1 is significantly more effective than ArSK/CCK2 when tested at concentrations of 25 and 100 nM (p < 0.01 and 0.05, respectively). Furthermore, ArSK/CCK2(ns) peptide did not cause contraction of tube foot preparations in vitro (data not shown).

Although ArSK/CCK1 expression was not detected in the apical muscle using immunohistochemistry with peroxidase-labelled secondary antibodies, double immunofluorescence labelling revealed the presence of both ArSK/CCK1 and ArSK/CCKR (see above). Accordingly, and consistent with the effects of ArSK/CCKP-derived neuropeptides on cardiac stomach and tube foot preparations, both ArSK/CCK1 and ArSK/CCK2 caused contraction of apical muscle preparations (*Figure 7e and f*). However, by comparison with the effect of acetylcholine, which was tested at concentration of 10 µM to normalise effects of the peptides on different preparations, the contracting actions of ArSK/CCK1 and ArSK/CCK2 were only ~40 % of the effect of 10 µM acetylcholine at the highest concentration tested (1 µM; *Figure 7f*). In contrast, the mean effects of ArSK/CCK1 and ArSK/CCK2 on tube foot preparations at 1 µM were ~220 % and ~110%, respectively, of the effect of 10 µM acetylcholine (*Figure 7d*). No significant differences in the effects of ArSK/CCK1 and ArSK/CCK2 on apical muscle preparations were observed (two-way ANOVA, p > 0.05) and, as was observed with tube feet, ArSK/CCK2(ns) had no effect on apical muscle preparations (data not shown).

## ArSK/CCK1 and ArSK/CCK2 trigger cardiac stomach retraction in vivo

Previous studies have revealed that the starfish neuropeptide NGFFYamide causes contraction of cardiac stomach preparations in vitro and triggers retraction of the everted cardiac stomach in vivo (*Semmens et al., 2013*). Because both ArSK/CCK1 and ArSK/CCK2 also cause contraction of cardiac stomach preparations in vitro, it was of interest to investigate if these neuropeptides also trigger retraction of the everted cardiac stomach in vivo. As reported previously (*Semmens et al., 2013*), cardiac stomach eversion was induced by immersing starfish in seawater containing 2 % added $MgCl_2$. In control experiments where starfish were injected with water, no retraction of the cardiac stomach was observed (*Figure 8*). However, injection of ArSK/CCK1 or ArSK/CCK2 (10 nmol; 10 µl of 1 mM) triggered cardiac stomach retraction (p < 0.0001 for both peptides; two-way ANOVA) (*Figure 8a, b and dFigure 8—videos 1; 2*), consistent with the contracting action of these peptides in vitro (*Figure 7a and b*). ArSK/CCK1 and ArSK/CCK2 triggered cardiac stomach retraction in all animals tested but with some variability in the rate and extent of retraction. Consistent with the modest effect of ArSK/CCK2(ns) on cardiac stomach preparations in vitro (*Figure 7b*), injection of ArSK/CCK2(ns) (10 nmol; 10 µl of 1 mM) did not trigger cardiac stomach retraction in vivo (*Figure 8c*).

## ArSK/CCK1 and ArSK/CCK2 inhibit feeding behaviour in *A. rubens*

The effects of ArSK/CCK1 and ArSK/CCK2 in triggering cardiac stomach retraction in *A. rubens* (see above) suggested that SK/CCK-type signalling may have a physiological role in inhibition and/or termination of feeding behaviour in starfish, which would be consistent with the physiological roles of SK/CCK-type neuropeptides in other taxa (*Al-Alkawi et al., 2017*; *Downer et al., 2007*; *Kang et al., 2011*; *Maestro et al., 2001*; *Meyering-Vos and Müller, 2007*; *Nachman et al., 1986b*; *Nässel et al., 2019*; *Rehfeld, 2017*; *Roman et al., 2017*; *Wei, 2000*; *Yu et al., 2013a*; *Zels et al., 2015*; *Zhang et al., 2017*). Therefore, we performed experiments to specifically investigate if ArSK/CCK1 and ArSK/CCK2 have inhibitory effects on starfish feeding behaviour on prey (mussels). Injection of ArSK/CCK1 or ArSK/CCK2 (10 nmol; 10 µl of 1 mM) did not affect the time taken for starfish to make first contact with a mussel (time to touch; *Figure 9a and b*). However, compared to water-injected controls, the mean time taken for starfish to adopt a feeding posture (time to enclose) was higher in neuropeptide-injected animals, reaching statistical significance (p < 0.05) with ArSK/CCK2 treatment (p < 0.05; *Figure 9d*) but showing only a tendency to an increased time to enclose with ArSK/CCK1

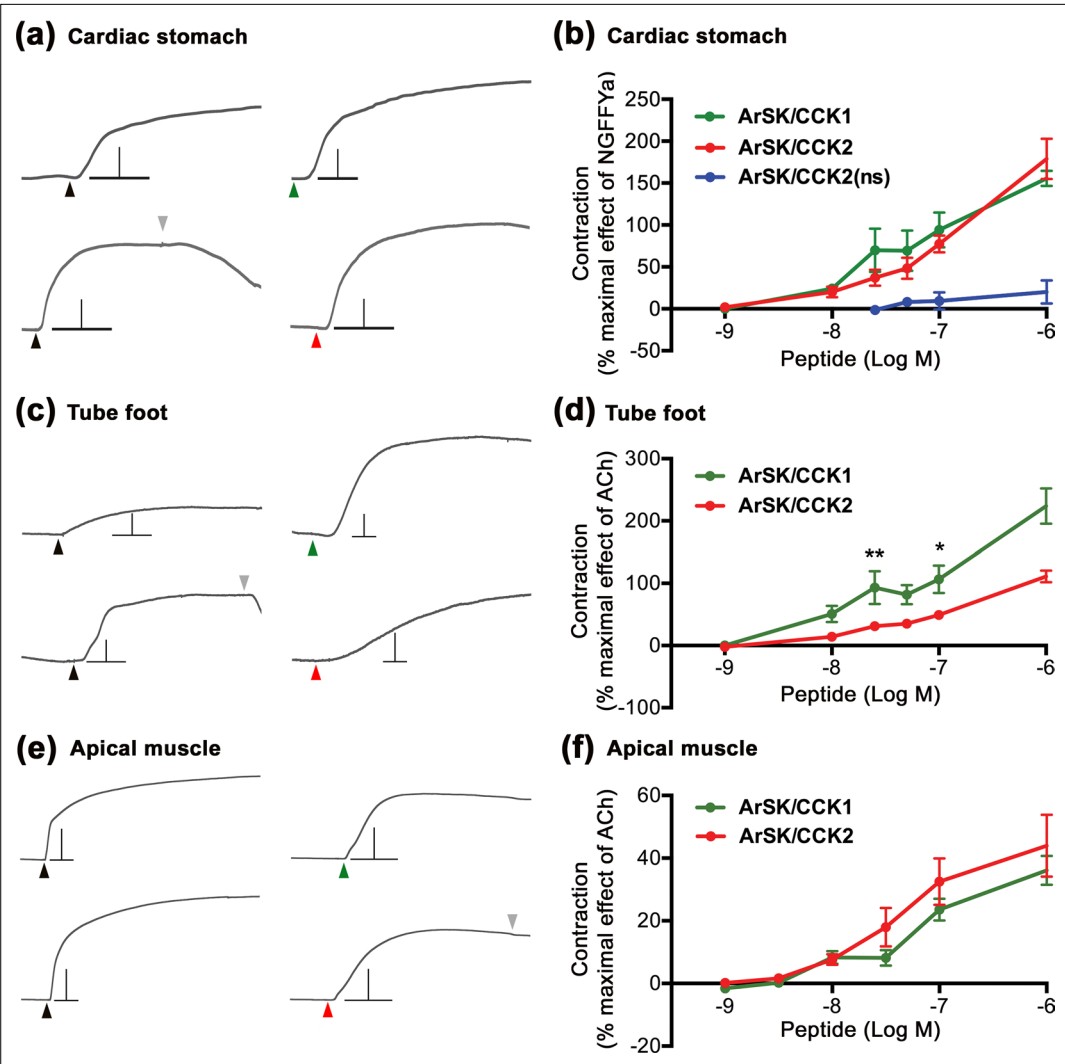

**Figure 7.** ArSK/CCK1 and ArSK/CCK2 cause concentration-dependent contraction of in vitro preparations of cardiac stomach, tube foot, and apical muscle from *Asterias rubens.* (**a**) Representative recordings of the effects ArSK/CCK1 (1 μM; green arrowhead) and ArSK/CCK2 (1 μM; red arrowhead) in causing contraction of cardiac stomach preparations and compared with the effect of NGFFYamide (100 nM; black arrowhead) on the same preparation. The downward pointing grey arrowhead shows when a preparation was washed with seawater. Scale bar: vertical 0.5 mV; horizontal 1 min. (**b**) Concentration-response curves comparing the effects of ArSK/CCK1, ArSK/CCK2, and ArSK/CCK2(ns) on cardiac stomach preparations. The effects of peptides (means ± s.e.m.; n = 5–9) were normalized to the effect of 100 nM NGFFYamide (NGFFYa). (**c**) Representative recordings of the effects ArSK/CCK1 (1 μM; green arrowhead) and ArSK/CCK2 (1 μM; red arrowhead) in causing contraction of tube foot preparations and compared with the effect of acetylcholine (10 μM; black arrowhead) on the same preparation. The downward pointing grey arrowhead shows when a preparation was washed with seawater. Scale bar: vertical 0.08 mV; horizontal 1 min. (**d**) Concentration-response curves comparing the effects of ArSK/CCK1 and ArSK/CCK2 on tube foot preparations. The effects of peptides (means ± s.e.m.; n = 8–10) were normalized to the effect of 10 μM acetylcholine (ACh). * indicates statistically significant differences between ArSK/CCK1 and ArSK/CCK2 when tested at concentrations of 25 and 100 nM (p < 0.01 and p < 0.05, respectively) as determined by two-way ANOVA and Bonferroni's multiple comparison test. (**e**) Representative recordings of the effects ArSK/CCK1 (1 μM; green arrowhead) and ArSK/CCK2 (1 μM; red arrowhead) in causing contraction of apical muscle preparations and compared to the effect of acetylcholine (10 μM; black arrowhead) on the same preparation. The downward pointing grey arrowhead shows when a preparation was washed with seawater. Scale bar: vertical 0.4 mV; horizontal 1 min. (**f**) Concentration-response curves comparing the effects of ArSK/CCK1 and ArSK/CCK2 on apical muscle preparations. The effects of peptides (means ± s.e.m; n = 20–23) were normalized to the effect of 10 μM acetylcholine (ACh).

*Figure 7 continued on next page*

*Figure 7 continued*

The online version of this article includes the following figure supplement(s) for figure 7:

**Source data 1.** Data for graphs shown in *Figure 7b, d and f*.

treatment (p = 0.0523; *Figure 9c*). This increased time to enclose was also reflected in an increased number of advances to touch the mussel in the starfish treated with ArSK/CCK1 or ArSK/CCK2 (data not shown). Moreover, by comparison with control starfish (water-injected), fewer starfish injected with ArSK/CCK1 or ArSK/CCK2 proceeded to initiation of a feeding posture after the first touch (p < 0.0001 and p < 0.01 for ArSK/CCK1 and ArSK/CCK2, respectively; *Figure 9e and f*). Another observation indicative of an inhibitory effect on feeding behaviour was that four and two of the starfish from the ArSK/CCK1- and ArSK/CCK2-treated groups, respectively, did not initiate feeding on a mussel within the 5 hr (300 min) observation period of the experiment, although feeding was commenced later and within 24 hr of initiating the experiment.

## Discussion

The evolutionary origin of SK/CCK-type neuropeptide signalling has been traced to the common ancestor of the Bilateria, informed by the molecular characterisation of the CCK/gastrin-type and SK-type neuropeptide signalling systems in chordates and protostomes, respectively (*Bloom et al., 2019*; *Janssen et al., 2008*; *Johnsen and Rehfeld, 1990*; *Kubiak et al., 2002*; *Mirabeau and Joly, 2013*; *Monstein et al., 1993*; *Nachman et al., 1986a*; *Schwartz et al., 2018*; *Yu et al., 2013b*; *Yu and Smagghe, 2014b*). Here, we report the first molecular and functional characterisation of an SK/CCK-type neuropeptide signalling system in a non-chordate deuterostome – the starfish *A. rubens* (phylum Echinodermata). This provides a key 'missing link' in our knowledge of the evolution and comparative physiology of SK/CCK-type signalling, complementing previously reported investigations of CCK/gastrin-type signalling in chordates and SK-type signalling in protostome invertebrates (e.g. insects).

### Molecular characterisation of SK/CCK-type signalling system in an echinoderm – the starfish *A. rubens*

Two SK/CCK-type neuropeptides (ArSK/CCK1 and ArSK/CCK2) derived from the precursor protein ArSK/CCKP were detected by mass spectrometry in *A. rubens* radial nerve cord extracts. An evolutionarily conserved feature of SK/CCK-type neuropeptides is a tyrosine (Y) residue that is post-translationally modified by the addition of a sulphate group (*Dufresne et al., 2006*; *Schwartz et al., 2018*; *Yu and Smagghe, 2014a*) and accordingly both ArSK/CCK1 and ArSK/CCK2 have a sulphated tyrosine. However, non-sulphated forms of these two peptides, ArSK/CCK1(ns) and ArSK/CCK2(ns), were also detected in *A. rubens* nerve cord extracts. Analysis of *A. rubens* neural transcriptome sequence data identified a GPCR (ArSK/CCKR) that is an ortholog of SK/CCK-type receptors that have been characterised in other taxa. Furthermore, heterologous expression of ArSK/CCKR in CHO-K1 cells revealed that the sulphated forms of ArSK/CCK1 and ArSK/CCK2 are potent agonists for ArSK/CCKR, whereas ArSK/CCK2(ns) exhibited little or no agonist activity on this receptor. Therefore, we conclude that the sulphated neuropeptides ArSK/CCK1 and ArSK/CCK2 and the GPCR ArSK/CCKR comprise the SK/CCK-type neuropeptide signalling system in the starfish *A. rubens*. The requirement for the tyrosine residues in ArSK/CCK1 and ArSK/CCK2 to be sulphated in order that these peptides can act as potent ligands for ArSK/CCK-R is consistent with the properties of many SK/CCK-type peptides in other taxa, including peptides that act as ligands for CCKR1 in mammals and SK-type peptides that act as ligands for the *Drosophila* receptor DSK-R1 (*Dufresne et al., 2006*; *Kubiak et al., 2002*). However, this is not a universal feature of SK/CCK-type signalling; for example, non-sulphated SK/CCK-type peptides act as potent ligands for CCK2R in mammals (*Dufresne et al., 2006*) and for CKR2a and CKR2b in the nematode *C. elegans* (*Janssen et al., 2008*).

Comparison of the sequences of ArSK/CCK1 and ArSK/CCK2 with the sequences of SK/CCK-type peptides that have been identified in other taxa reveal several evolutionarily conserved features, including C-terminal amidation and the aforementioned tyrosine (Y) residue. The position of the tyrosine (Y) residue with respect to the C-terminal amide is variable amongst SK/CCK-type peptides, with

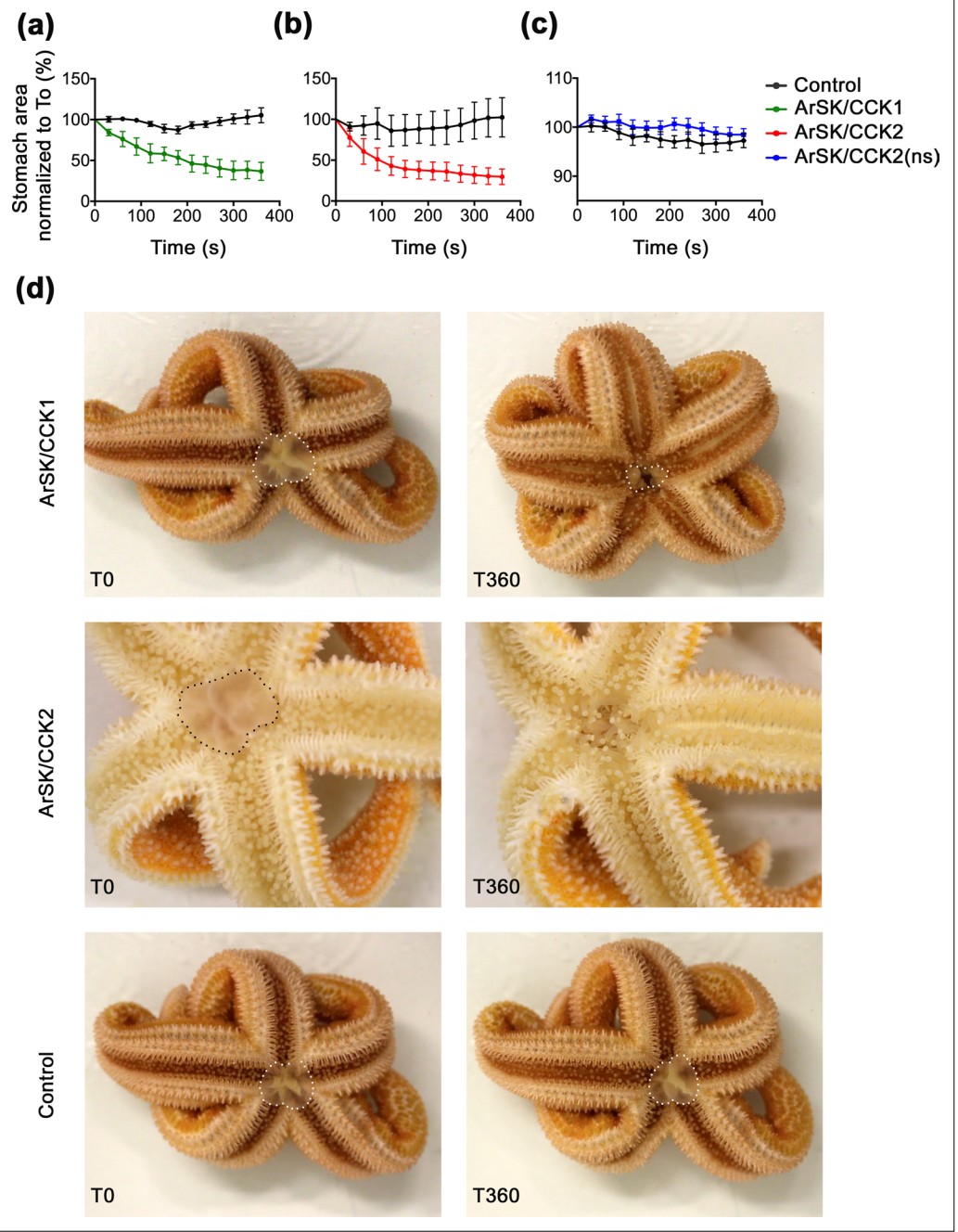

**Figure 8.** ArSK/CCK1 and ArSK/CCK2 trigger cardiac stomach retraction in *Asterias rubens*. The graphs compare experiments where starfish were first injected with vehicle (black line; 10 µl of distilled water) and then injected with (**a**) ArSK/CCK1 (green line; 10 mol; 10 µl 1 mM), or (**b**) ArSK/CCK2 (red line; 10 mol; 10 µl 1 mM), or (**c**) ArSK/CCK2(ns) (blue line; 10 mol; 10 µl 1 mM). Stomach eversion was induced by placing starfish in seawater containing 2 % MgCl₂ and then the area of cardiac stomach everted (in 2D) at 30 s intervals (0–360 s) following injection of water (control) or peptide was measured, normalizing to the area of cardiac stomach everted at the time of injection (T0). Data (means ± s.e.m.) were obtained from 6 (ArSK/CCK1), 6 (ArSK/CCK2), or 8 (ArSK/CCK2(ns)) experiments. Both ArSK/CCK1 and ArSK/CCK2 cause retraction of the cardiac stomach, with >50% reduction in the area of cardiac stomach everted within 360 s (*Figure 8—videos 1; 2*), whereas ArSK/CCK2(ns) has no effect. (**d**) Photographs from representative experiments showing that injection of ArSK/CCK1 (10 mol; 10 µl 1 mM at T0) or ArSK/CCK2 (10 mol; 10 µl 1 mM at T0) causes retraction of the everted cardiac stomach (marked with white or black dots), which is reflected in a reduction in the area everted after 360 s (T360). By way of comparison, in a control experiment injection with vehicle (10 µl of distilled water at T0) does not trigger cardiac stomach retraction.

*Figure 8 continued on next page*

*Figure 8 continued*

See *Figure 8—source data 1* for source data.

The online version of this article includes the following video and figure supplement(s) for figure 8:

**Source data 1.** Data for graphs shown in *Figure 8a, b and c,*.

**Figure 8—video 1.** ArSK/CCK1 (10 mol; 10 µl 1 mM) induced retraction of the cardiac stomach in the starfish *Asterias rubens*.

https://elifesciences.org/articles/65667/figures#fig8video1

**Figure 8—video 2.** ArSK/CCK2 (10 mol; 10 µl 1 mM) induced retraction of the cardiac stomach in the starfish *Asterias rubens*.

https://elifesciences.org/articles/65667/figures#fig8video2

between five and seven intervening residues. Accordingly, in ArSK/CCK1, ArSK/CCK2, and SK/CCK-type peptides from other echinoderms and hemichordates (collectively Ambulacraria), there are six intervening residues. In ArSK/CCK1 the tyrosine (Y) residue is preceded by an aspartate (D) residue and the tyrosine (Y) residue is followed by a glycine (G) residue; accordingly, a DYG motif is also a feature of many SK/CCK-type peptides in lophotrochozoans and arthropods, suggesting that this may be an ancestral characteristic that has been lost in some bilaterian taxa (e.g. chordates, nematodes). ArSK/CCK1, but not ArSK/CCK2, has an N-terminal pyroglutamate in its mature form and this post-translational modification is also predicted or known to occur in some SK/CCK-type peptides in other taxa – for example, the SK1 and SK2 peptides from the cockroach *L. maderae* (*Predel et al., 1999*) and the SK/CCK-type peptides in the molluscs *Pinctata fucata* and *C. gigas* (*Schwartz et al., 2018*; *Stewart et al., 2014*). The C-terminal residue of ArSK/CCK2 is a phenylalanine residue (F) and in this respect ArSK/CCK2 is like most SK/CCK-type peptides that have been identified in other taxa. It is noteworthy, therefore, that the C-terminal residue of ArSK/CCK1 is a tryptophan (W) residue, which is also a feature of an SK/CCK1-like peptide in another starfish species – the crown-of-thorns starfish *Acanthaster planci* (*Smith et al., 2017*). Furthermore, this unusual feature of one of the two SK/CCK-type peptides that occur in starfish species appears to be unique to this class of echinoderms (the Asteroidea) because SK/CCK-type peptides that have been identified in other echinoderms all have the more typical C-terminal phenylalanine (F) residue (*Chen et al., 2019*; *Zandawala et al., 2017*). Interestingly, however, it is not completely unique to starfish because in the bivalve mollusc *C. gigas* there are two SK/CCK-type peptides, one of which has a C-terminal phenylalanine (F) and another that has a tryptophan (W) residue (*Schwartz et al., 2018*). Thus, it appears that SK/CCK-type neuropeptides with a C-terminal tryptophan (W) residue evolved independently in starfish and in the bivalve mollusc *C. gigas*.

## Functional characterization of SK/CCK-type neuropeptides in an echinoderm – the starfish *A. rubens*

Having identified the molecular components of an SK/CCK-type neuropeptide signalling system in *A. rubens*, comprising the sulphated neuropeptides ArSK/CCK1 and ArSK/CCK2 and their receptor ArSK/CCKR, our next objective was to gain insights into the physiological roles of SK/CCK-type neuropeptides in starfish. Using mRNA in situ hybridisation, we examined the anatomical expression patterns of ArSK/CCKP transcripts and using immunohistochemistry we analysed and compared the anatomical expression patterns of ArSK/CCK1 and ArSK/CCKR. This revealed a widespread pattern of expression, including the CNS, tube feet, and other body wall associated structures and the digestive system, as discussed below. This pattern of expression can be interpreted with reference to the anatomy of the starfish nervous system (*Cobb, 1970*; *Mashanov et al., 2016*; *Smith, 1937*), digestive system (*Anderson, 1953*; *Anderson, 1954*), and body wall (*Blowes et al., 2017*).

The starfish CNS consists of radial nerve cords and a circumoral nerve ring, where cells expressing ArSK/CCKP and ArSK/CCK1 were revealed in both the ectoneural and hyponeural regions. The presence of an extensive network of immunostained fibres in the neuropile of the ectoneural region is consistent with neuronal expression of ArSK/CCKP and ArSK/CCK1. Furthermore, regional variation in the density of ArSK/CCK1-ir fibres in the ectoneural neuropile was observed. The ectoneural region is thought to largely contain sensory neurons and interneurons (*Mashanov et al., 2016*; *Cobb, 1970*; *Smith, 1937*), but the functional identity of neuronal cell types and the neural circuitry are not known.

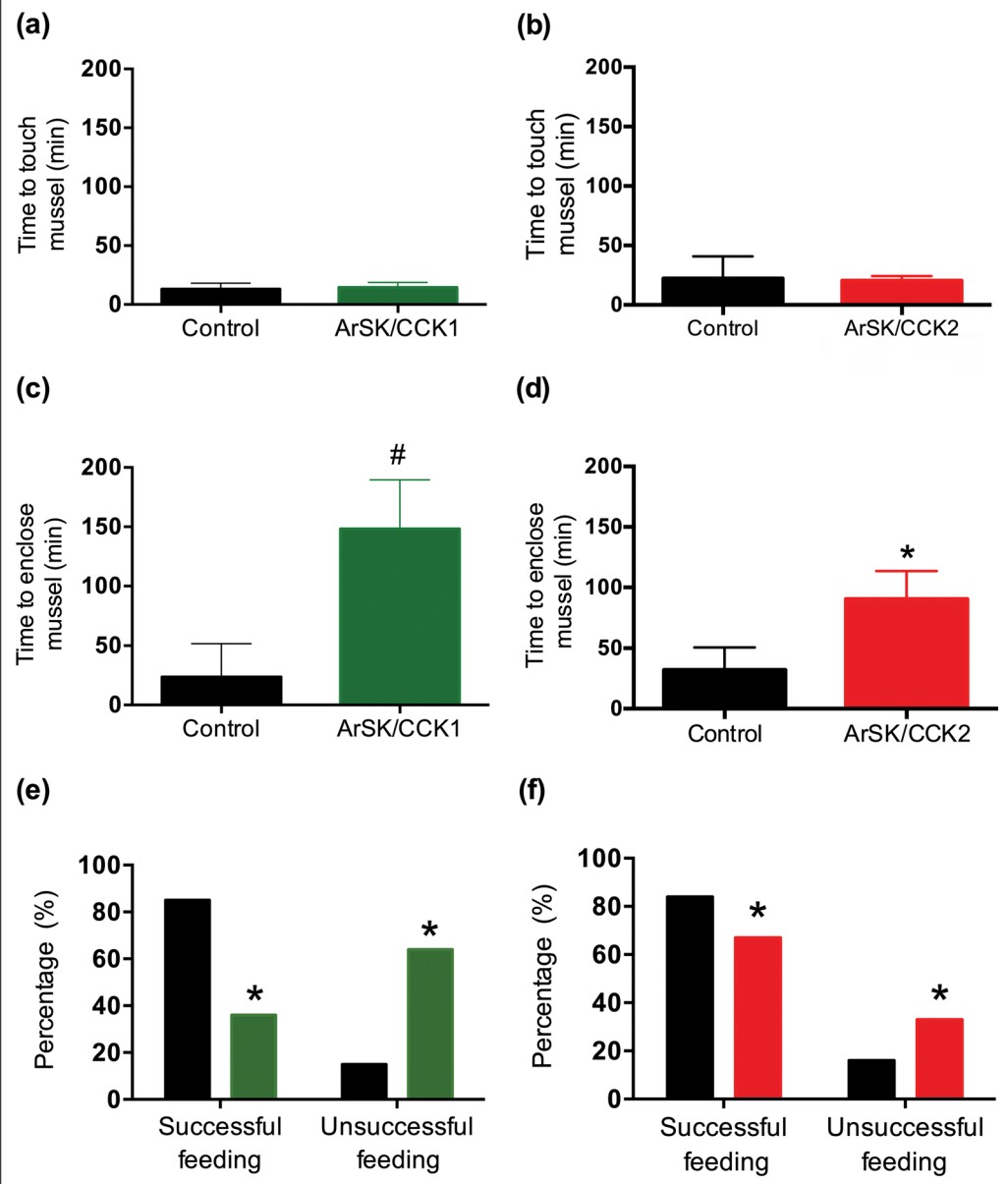

**Figure 9.** Effects of ArSK/CCK1 and ArSK/CCK2 on feeding behaviour in *Asterias rubens*. To investigate if injection of ArSK/CCK1 and/or ArSK/CCK2 affects feeding behaviour, starved animals were presented with a mussel as prey and then behaviour was observed. By comparison with vehicle-injected animals (control; 10 µl of distilled water; shown in black), injection of ArSK/CCK1 (**a**; 10 mol or 10 µl of 1 mM; shown in green) or ArSK/CCK2 (**b**; 10 mol or 10 µl of 1 mM; shown in red) had no effect on the time taken for starfish to make first contact with the mussel. However, by comparison with vehicle-injected animals (control; 10 µl of distilled water; shown in black) injection of ArSK/CCK1 (**c**; 10 mol or 10 µl of 1 mM; shown in green) or ArSK/CCK2 (**d**; 10 mol or 10 µl of 1 mM; shown in red) causes an increase in the time elapsed before starfish enclose a mussel. Data are expressed as means ± s.e.m. (n = 13 for control- and 11 for ArSK/CCK1-treated groups; n = 19 for control- and 19 for ArSK/CCK2-treated groups). # indicates a nearly statistically significant difference (p = 0.0523) between vehicle-injected and ArSK/CCK1-injected groups, as determined by two-tailed Mann-Whitney U-test. * indicates statistically significant differences (p < 0.05) between vehicle-injected and ArSK/CCK2-injected groups, as determined by two-tailed Welch's unequal variances t-test. Furthermore, injection of ArSK/CCK1 (**e**; 10 mol or 10 µl of 1 mM; shown in green) or ArSK/CCK2 (**f**; 10 mol or 10 µl of 1 mM; shown in red) causes a significant decrease in the percentage of starfish that initiate feeding after the mussel is touched for the first time, by comparison with vehicle-injected animals (control; 10 µl of distilled water; shown in black). * indicates statistically significant differences (p < 0.0001 and p < 0.01 for ArSK/CCK1- and ArSK/CCK2-treated groups, respectively) between vehicle-injected and ArSK/CCK1- or ArSK/CCK2-injected

*Figure 9 continued on next page*

*Figure 9 continued*

groups, as determined by Fisher's exact test. See *Figure 9—source data 1* for source data.

The online version of this article includes the following figure supplement(s) for figure 9:

**Source data 1.** Data for graphs shown in *Figure 9*.

Therefore, it is not possible at present to determine the functional properties of neurons expressing the ArSK/CCKP gene in the ectoneural region of the CNS. However, comparative immunohistochemical analysis of ArSK/CCK1 and ArSK/CCKR expression revealed a 'salt and pepper' pattern of labelling of fibres in the ectoneural neuropile, indicative of intercellular SK/CCK-type peptidergic signalling in this region of the CNS.

The hyponeural region of the starfish CNS only comprises motoneurons and the projection pathways of the axons of these motoneurons have been reported (*Lin et al., 2017a*; *Smith, 1950*). Thus, the axons of hyponeural motor neurons project around the tube feet and coalesce as a fibre bundle known as the lateral motor nerve (*Lin et al., 2017a*; *Smith, 1937*). Consistent, with the expression of ArSK/CCKP in hyponeural cells, ArSK/CCK1-ir fibres can be seen in the lateral motor nerves. Branches of the lateral motor nerves project into the coelomic lining of the body wall in starfish (*Smith, 1937*; *Smith, 1950*) and accordingly ArSK/CCK1-ir fibres were observed in coelomic lining of the body wall in *A. rubens*. Furthermore, the presence of ArSK/CCK1-ir fibres in inter-ossicular tissue of the body wall in *A. rubens* suggests that ArSK/CCKP-expressing hyponeural cells may include motoneurons that innervate inter-ossicular muscles and/or inter-ossicular mutable collagenous tissue (*Blowes et al., 2017*). Interestingly, double immunofluorescence labelling indicates that there are distinct clusters of hyponeural motoneurons that express ArSK/CCK1 or ArSK/CCKR, although ArSK/CCKR appears to be co-expressed with ArSK/CCK1 in some hyponeural motoneurons. The functional significance of these findings is at present unknown, but they can be interpreted in the context of the complex patterns of neuropeptide and neuropeptide receptor expression/co-expression that have been reported in the CNS of other taxa (*Smith et al., 2019*).

ArSK/CCKP and ArSK/CCK1 are widely expressed in the digestive system of *A. rubens*, including the oesophagus, peristomial membrane, cardiac stomach, pyloric stomach, pyloric duct, pyloric caeca, intestine, and rectal caeca. Furthermore, in vitro pharmacological experiments revealed that ArSK/CCK1 and ArSK/CCK2 cause concentration-dependent contraction of cardiac stomach preparations. ArSK/CCKP and ArSK/CCK1 expressing cells are present in the mucosal wall of the cardiac stomach and a dense network of ArSK/CCK1-ir fibres is present in the basiepithelial nerve plexus of the cardiac stomach, particularly in the highly folded lateral pouches of the cardiac stomach. Accordingly, analysis of ArSK/CCKR expression revealed immunostaining in the basiepithelial nerve plexus and visceral muscle layer of the pyloric and cardiac stomach and in the nodule that links the cardiac stomach to extrinsic retractor strands. Therefore, SK/CCK-type neuropeptides released by the axonal processes of mucosal neurons located in the basiepithelial nerve plexus may cause stomach contraction by acting directly on visceral muscle and/or indirectly by triggering the release of other myoactive substances from the basiepithelial nerve plexus.

Starfish exhibit one of the most remarkable feeding behaviours in the animal kingdom – they evert their stomach out of their mouth and digest large prey externally, and once the prey has been digested, the stomach is withdrawn. Extra-oral feeding in starfish requires relaxation of muscle in the wall of the cardiac stomach and in intrinsic and extrinsic retractor strands to enable stomach eversion. Then when external digestion and ingestion of prey tissue is completed, contraction of the musculature enables cardiac stomach retraction (*Anderson, 1954*). Previous studies have identified neuropeptides that cause relaxation of the cardiac stomach in vitro and trigger cardiac stomach eversion when injected in vivo; for example, the SALMFamide neuropeptide S2 (*Melarange et al., 1999*) and the vasopressin/oxytocin-type neuropeptide asterotocin (*Odekunle et al., 2019*). Conversely, NGFFYamide has been identified as a neuropeptide in *A. rubens* that triggers contraction of the cardiac stomach in vitro and retraction of the everted stomach when injected in vivo (*Semmens et al., 2013*). The in vitro effects of ArSK/CCK1 and ArSK/CCK2 in causing contraction of the cardiac stomach and the presence of ArSK/CCK1-ir and ArSK/CCKR-ir in the cardiac stomach, nodule, and retractor strands suggested that SK/CCK-type peptides may participate in mechanisms of cardiac stomach retraction physiologically. Accordingly, injection of 10 nmol of ArSK/CCK1 or ArSK/CCK2 triggered partial or complete

retraction of the cardiac stomach within a test period of 6 min. These experiments indicate that SK/CCK-type peptides may be involved in physiological mechanisms of cardiac stomach retraction in starfish. Furthermore, to investigate the importance of tyrosine (Y) sulphation for the bioactivity of SK/CCK-type peptides in *A. rubens*, we also tested the non-sulphated peptide ArSK/CCK2(ns) on cardiac stomach preparations. By comparison with ArSK/CCK1 and ArSK/CCK2, ArSK/CCK2(ns) had a modest contracting effect on cardiac stomach preparations in vitro and did not trigger cardiac stomach retraction in vivo. This is consistent with the low potency of ArSK/CCK2(ns) as an agonist on ArSK/CCKR expressed in CHO cells and indicates that non-sulphated SK/CCK-type peptides are not bioactive physiologically in starfish.

The in vivo effect of SK/CCK-type neuropeptides in triggering cardiac stomach retraction is indicative of a role in physiological mechanisms that control termination of feeding behaviour in starfish. By way of comparison, the neuropeptide NGFFYamide that triggers retraction of the everted stomach of *A. rubens* also causes a significant delay in the onset of feeding on prey (mussels) when injected in vivo (*Tinoco et al., 2018*). Accordingly, here we observed that starfish injected with SK/CCK-type neuropeptides took longer to enclose prey compared to control animals injected with water. Furthermore, in animals injected with SK/CCK-type neuropeptides the proportion of starfish that successfully consumed prey was fewer than in control animals that were injected with water. Thus, collectively these findings indicate that SK/CCK-type signalling acts as a physiological regulator that inhibits and/or terminates feeding behaviour in starfish. Further investigation of how SK/CCK-type signalling regulates feeding-related processes in starfish may be facilitated by use of gene-knockdown methods (e.g. RNA interference; *Alicea-Delgado et al., 2021*).

Lastly, informed by detection of ArSK/CCKP, ArSK/CCK1, and ArSK/CCKR expression in the tube feet and apical muscle of *A. rubens*, we also tested ArSK/CCK1 and ArSK/CCK2 on in vitro preparations of these organs/tissues and found that both peptides cause concentration-dependent contraction. Thus, SK/CCK-type neuropeptides appear to have a general role as myoexcitatory agents in starfish.

## Comparative and evolutionary physiology of SK/CCK-type neuropeptide signalling in the Bilateria

The myoexcitatory effects of SK/CCK-type neuropeptides in *A. rubens* are consistent with findings from previous studies on vertebrates and protostome invertebrates. For example, CCK causes pyloric and gall bladder contraction in mammals (*Gutiérrez et al., 1974*; *Rehfeld, 2017*; *Shaw and Jones, 1978*; *Vizi et al., 1973*) and gut contraction in other vertebrates (*Tinoco et al., 2015*). Likewise, SK-type peptides also have myoexcitatory effects in insects (*Al-Alkawi et al., 2017*; *Maestro et al., 2001*; *Marciniak et al., 2011*; *Nachman et al., 1986b*; *Nichols, 2007*; *Palmer et al., 2007*; *Predel et al., 2001*). However, SK/CCK-type peptides are not exclusively myoexcitatory because, for example, CCK causes relaxation of the proximal stomach in mammals; however, this myoinhibitory effect of CCK is indirect and mediated by vagal and splanchnic afferents (*Takahashi and Owyang, 1999*). Accordingly, both inhibitory and excitatory effects of CCK-8 on the stomach of a non-mammalian vertebrate, the rainbow trout *Oncorhynchus mykiss*, have been reported (*Olsson et al., 1999*) and an SK/CCK-type peptide causes a decrease in the frequency of hindgut contraction in the mollusc *C. gigas* (*Schwartz et al., 2018*). Furthermore, and directly relevant to this study, it has been reported that mammalian CCK-8 causes in vitro relaxation of intestine preparations from the sea cucumber of *Holothuria glaberrima* (*García-Arrarás et al., 1991*). With the determination of the amino acid sequences of native SK/CCK-type neuropeptides in sea cucumbers (*Chen et al., 2019*; *Zandawala et al., 2017*), it will now be possible to specifically investigate their pharmacological effects in these animals to make direct comparisons with the findings reported here for starfish.

As discussed below, SK/CCK-type neuropeptides are perhaps best known for their roles as inhibitory regulators of feeding. However, in common with other neuropeptides, they are pleiotropic in their physiological roles. Thus, linked to regulation feeding, SK/CCK-type neuropeptides stimulate secretion of gastric acid and/or digestive enzymes in mammals, insects, nematodes, ascidians, and molluscs (*Bevis and Thorndyke, 1981*; *Chen et al., 2004*; *Harper and Raper, 1943*; *Harshini et al., 2002b*; *Janssen et al., 2008*; *Nachman et al., 1997*; *Shaw and Jones, 1978*; *Thorndyke and Bevis, 1984*; *Zels et al., 2015*). Accordingly, expression of SK/CCK-type peptides by cells in several regions of the digestive system in *A. rubens* may be indicative of a similar role in starfish. Furthermore, CCK precursor transcripts are detected in rat spinal motoneurons (*Cortés et al., 1990*) and SK-type neuropeptides act as positive growth regulators for neuromuscular junction formation and promote locomotion in larval *Drosophila*

(*Chen and Ganetzky, 2012*). In this context, it is noteworthy that SK/CCK-type neuropeptides cause contraction of body wall-associated muscles (apical muscle) and organs (tube feet) in starfish. Accordingly, the expression of SK/CCK-type peptides in hyponeural motoneurons, the lateral motor nerve, and inter-ossicular tissue of the body wall of *A. rubens* may reflect evolutionarily ancient and conserved roles of SK/CCK-type neuropeptides as regulators of skeletal muscle function. It is also noteworthy that CCK is one of the most abundantly expressed neuropeptides in the cortex of the mammalian brain, where it is expressed by sub-populations of GABAergic interneurons and acts as a multi-functional molecular switch to regulate the output of cortical neuronal circuits (*Lee and Soltesz, 2011*). Furthermore, evidence that expression of CCK in GABAergic neurons is an evolutionarily ancient association was provided by a recent study reporting co-localisation of CCK-8 and GABA in several different neuronal populations in the brain of the sea lamprey *Petromyzon marinus* (*Sobrido-Cameán et al., 2020*). GABA-immunoreactive neurons have been revealed in the ectoneural region of the radial nerve cord in *A. rubens* (*Newman and Thorndyke, 1994*) and sub-populations of these neurons may correspond with cells expressing SK/CCK-type neuropeptides reported in this study.

The key functional insights from this study are our observations that in the starfish *A. rubens* SK/CCK-type neuropeptides trigger cardiac stomach contraction and retraction and induce a delay in the onset of feeding and a reduction in predation. These findings are of general interest because of the previously reported evidence that SK/CCK-type neuropeptides mediate physiological mechanisms of satiety and/or regulate feeding behaviour in vertebrates, insects, and the mollusc *C. gigas* (*Al-Alkawi et al., 2017*; *Downer et al., 2007*; *Himick and Peter, 1994*; *Kang et al., 2011*; *Maestro et al., 2001*; *Meyering-Vos and Müller, 2007*; *Nachman et al., 1986a*; *Nachman et al., 1986b*; *Nässel and Zandawala, 2019*; *Rehfeld, 2017*; *Roman et al., 2017*; *Schwartz et al., 2018*; *Wei, 2000*; *Yu et al., 2013a*; *Yu and Smagghe, 2014b*; *Zels et al., 2015*; *Zhang et al., 2017*). Furthermore, insights into the mechanisms by which SK/CCK-type neuropeptides regulate feeding behaviour in mammals and insects have been obtained. In mammals CCK released by intestinal endocrine cells acts on vagal afferents, which is thought to then lead to activation of calcitonin gene-related peptide-expressing neurons in the parabrachial nucleus that supress feeding and inhibition of Agouti-related peptide-expressing hypothalamic neurons that promote feeding (*Beutler et al., 2017*; *Essner et al., 2017*). In *Drosophila*, SK-type neuropeptides are expressed by a sub-population median neurosecretory cells in the brain that also produce insulin-like peptides and results from a variety of experimental studies indicate that release of SK-type neuropeptides by these neurons induces satiety in both larval and adult flies (*Nässel and Williams, 2014*). Thus, an evolutionarily conserved physiological role of SK/CCK-type neuropeptides as inhibitory regulators of feeding are mediated by different mechanisms in mammals and insects, which may reflect evolutionary divergence in the anatomy of these taxa. It is interesting, therefore, that in the unique context of the evolutionary and developmental replacement of bilateral symmetry with pentaradial symmetry in adult echinoderms, an ancient role of SK/CCK-type neuropeptides as inhibitory regulators of feeding-related processes has been retained in starfish. Furthermore, because feeding in starfish is accomplished by stomach eversion, there is a direct link between the action of SK/CCK-type neuropeptides on the gastro-intestinal system and inhibition/termination of feeding behaviour. Thus, our findings from starfish reported here for SK/CCK-type neuropeptides and previously for other neuropeptides (*Cai et al., 2018*; *Odekunle et al., 2019*; *Tian et al., 2017*; *Tinoco et al., 2018*; *Zhang et al., 2020*) reveal how ancient roles of neuropeptide signalling systems have been preserved in spite of unique and radical evolutionary and developmental changes in the anatomy of echinoderms amongst bilaterian animals.

## Materials and methods

**Key resources table**

| Reagent type (species) or resource | Designation | Source or reference | Identifiers | Additional information |
|---|---|---|---|---|
| Sequence-based reagent | 5'-TCGCTACTGTTTCTCTCGCA-3' 5'-AAAGGCGTCAACAACTGCTT-3' | Custom synthesized by Sigma-Aldrich | | Oligonucleotide primers for cloning of ArSK/CCKP cDNA (*Asterias rubens*) |
| Recombinant DNA reagent | Zero Blunt Topo | Thermo Fisher Scientific | Cat. no. 450,159 | |

*Continued on next page*

*Continued*

| Reagent type (species) or resource | Designation | Source or reference | Identifiers | Additional information |
|---|---|---|---|---|
| Recombinant DNA reagent | pBluescript SKII (+) | Agilent Technologies | Cat. no. 212,205 | |
| Recombinant DNA reagent | pcDNA 3.1+ vector with neomycin selectable marker (mammalian expression vector) | Invitrogen | Cat. no. V790-20 | |
| Transfected construct (*Asterias rubens*) | *Asterias rubens* ArSK/CCKR cDNA cloned in expression vector pcDNA 3.1+ | This paper | GenBank: MW261740 | See Materials and methods section 'Cell lines and pharmacological characterization of ArSK/CCKR' |
| Cell line (*Cricetus griseus*) | Chinese hamster ovary cells (CHO-K1) | Sigma-Aldrich | RRID: CVCL_0214 | Cat. no. 85051005 |
| Antibody | (Rabbit polyclonal) antibodies to ArSK/CCK1 | This paper | RRID: AB_2877176 | Antibodies generated by Elphick group at QMUL (m.r.elphick@qmul.ac.uk) – see Materials and methods section 'Generation and characterisation of antibodies to ArSK/CCK1 and ArSK/CCKR'. Affinity-purified antibodies were used for immunohistochemistry (1:10 or 1:15). |
| Antibody | (Guinea pig polyclonal) antibodies to ArSK/CCKR | This paper | RRID: AB_2891354 | Antibodies generated by Elphick group at QMUL (m.r.elphick@qmul.ac.uk) – see Materials and methods section 'Generation and characterisation of antibodies to ArSK/CCK1 and ArSK/CCKR'. Affinity-purified antibodies were used for immunohistochemistry (1:3.3). |
| Antibody | (Sheep polyclonal) Sheep alkaline phosphatase (AP)-conjugated anti-DIG antibody | Roche | Cat. no. 11093274910 | Used for mRNA in situ hybridisation (1:3000). |
| Antibody | (Goat polyclonal) Peroxidase-AffiniPure Goat Anti-Rabbit IgG (H + L) Horseradish Peroxidase conjugated | Jackson ImmunoResearch | RRID: AB_2313567 Cat. no. 111-035-003 | Used for immunohistochemistry (1:1000). |
| Antibody | (Goat polyclonal) Alexa Fluor 488-conjugated AffiniPure Goat Anti-Rabbit IgG (H + L) | Jackson ImmunoResearch | RRID: AB_2338046 Cat. no. 111-545-003 | Used for immunohistochemistry (1:500). |
| Antibody | (Goat polyclonal) Cy3-conjugated AffiniPure Goat Anti-Guinea Pig IgG (H + L) | Jackson ImmunoResearch | RRID: AB_2337423 Cat. no. 106-165-003 | Used for immunohistochemistry (1:300). |
| Commercial assay, kit | AminoLink Plus Immobilization Kit | Thermo Fisher Scientific | Cat. no. 44,894 | |
| Commercial assay, kit | MycoAlert Mycoplasma Detection Kit | Lonza | LT07-118 | |
| Software, algorithm | MSConvert | ProteoWizard Toolkit | Version 3.0.5759 | doi: 10.1093/bioinformatics/btn323 |
| Software, algorithm | Scaffold | Proteome Software Inc | Version 4.6.1 RRID:SCR_014345 | |
| Software, algorithm | MAFFT | MAFFT | Version 7 RRID:SCR_011811 | http://mafft.cbrc.jp/alignment/server/ |
| Software, algorithm | IQ-tree web server | http://iqtree.cibiv.univie.ac.at | | |
| Software, algorithm | MUSCLE | https://www.ebi.ac.uk/Tools/msa/muscle/ | RRID:SCR_011812 | |
| Software, algorithm | Volocity | PerkinElmer | Version 6.3.1 | |
| Software, algorithm | Adobe Photoshop CC | Adobe | Version 19.1.4, ×64 RRID:SCR_014199 | |
| Software, algorithm | Leica Application Suite Advanced Fluorescence (LAS AF) | Leica | Version 2.6.0.7266 | |
| Software, algorithm | LabChart | ADInstruments | Version 8.0.7 RRID:SCR_017551 | |
| Software, algorithm | ImageJ | http://rsb.info.nih.gov/ij | Version 1.0 RRID:SCR_003070 | |
| Software, algorithm | Prism | GraphPad | Version 6.0 RRID:SCR_002798 | |

## Animals

Adult starfish (*A. rubens*, Linnaeus, 1758) were collected at low tide near Margate (Kent, UK) or obtained from a fisherman based at Whitstable (Kent, UK). The starfish were maintained in a circulating seawater aquarium under a 12 hr–12 hr light-dark cycle (lights on at 8 a.m.) at a temperature of ~12 °C and salinity of 32‰, located in the School of Biological and Behavioural Sciences at Queen Mary University of London. Animals were fed on mussels (*Mytilus edulis*) that were collected at low tide near Margate (Kent, UK). Additionally, juvenile specimens of *A. rubens* (diameter 0.5–1.5 cm) used for anatomical studies were collected at the University of Gothenburg Sven Lovén Centre for Marine Infrastructure (Kristineberg, Sweden).

## Cloning of a cDNA encoding the *A. rubens* SK/CCK-type precursor ArSK/CCKP

A cDNA encoding the ArSK/CCK precursor (ArSK/CCKP), including 5' and 3' untranslated regions and the complete open reading frame, was amplified by PCR (Phusion High-Fidelity PCR Master Mix, NEB, Hitchin, Hertfordshire, UK) using specific oligonucleotide primers (5'-TCGCTACTGTTTCTCTCGCA-3' and 5'-AAAGGCGTCAACAACTGCTT-3'), which were designed using Primer3 software (http://bioinfo.ut.ee/primer3-0.4.0/) with reference to the ArSK/CCKP transcript sequence (contig 1124413; GenBank accession number KT601716) obtained from *A. rubens* radial nerve cord transcriptome data (*Semmens et al., 2016*). The PCR product was gel-extracted and purified (QIAquick Gel Extraction Kit, Qiagen, Manchester, UK) before being blunt-end cloned into pBluescript SKII (C) (Agilent Technologies, Stockport, Cheshire, UK) or Zero Blunt Topo PCR (Thermo Fisher Scientific, Waltham, MA) vectors. The clones were sequenced (Eurofins Genomics GmbH, Ebersberg, Germany) using the T7 and T3 sequencing primer sites.

## Localisation of ArSK/CCKP expression in *A. rubens* using mRNA in situ hybridization

To enable visualisation of ArSK/CCKP transcripts in *A. rubens* using mRNA in situ hybridization, digoxygenin-labelled RNA probes were synthesised. Zero Blunt Topo or pBluescript SKII (+) vectors containing the ArSK/CCKP cDNA were purified (Qiagen Maxiprep, Qiagen, Manchester, UK) and 5 µg of the vector was linearized using restriction enzymes (NEB, Hitchin, Hertfordshire, UK). Linearized vector containing the ArSK/CCKP cDNA were cleaned using phenol-chloroform (Sigma-Aldrich, Gillingham, UK) and chloroform-isomylalcohol (Sigma-Aldrich, Gillingham, UK) extractions and then precipitated using 0.1 volume of 3 M sodium acetate and 2.5 volume of 100 % ethanol (Honeywell, Fisher Scientific UK Ltd, Loughborough, UK) at –80 °C. The pellet was washed with 70 % ice-cold ethanol before air drying and re-suspending in autoclaved water. Sense and antisense RNA probes were synthesized using digoxigenin nucleotide triphosphate (DIG-NTP) mix (Roche, Mannheim, Germany), 5 × transcription buffer (NEB, Hitchin, Hertfordshire, UK), 0.2 M dithiothreitol (Promega, Madison, WI), placental ribonuclease inhibitor (10 U/µl) (Promega, Madison, WI), and T7 polymerase (50 U/µl) or T3 polymerase (50 U/µl) (NEB, Hitchin, Hertfordshire, UK) with 1 µg of linearised vector containing the ArSK/CCKP cDNA. Template DNA was then digested with RNase free DNase (NEB, Hitchin, Hertfordshire, UK). RNA probes were stored in 25 % formamide/2 × saline-sodium citrate (25 % FA/2 × SSC; VWR Chemicals, Leicestershire, UK) at –20 °C for long-term storage.

To prepare specimens of *A. rubens* for mRNA in situ hybridisation, animals were fixed by immersion in 4 % paraformaldehyde (PFA; Sigma-Aldrich, Gillingham, UK) in phosphate-buffered saline (PBS) overnight at 4 °C. Specimens were washed in PBS, dissected and placed in Morse's solution (10 % sodium citrate; 20 % formic acid in autoclaved water) to enable decalcification of ossicles in the body wall of starfish. Decalcified specimens were then washed in autoclaved water, dehydrated through a graded ethanol series, and then immersed in xylene (Honeywell, Fisher Scientific UK Ltd, Loughborough, UK) before being embedded in paraffin wax; 14 µm sections of *A. rubens* arms and central disk were prepared using an RM 2145 microtome (Leica Microsystems [UK], Milton Keynes, UK). Sections were collected on poly-L-lysine coated slides (VWR Chemicals, Lutterworth, Leicestershire, UK) that had been placed on a hot plate and covered with autoclaved water. Slides were left to dry before proceeding with probe hybridization and immunodetection.

Slides were kept at 60 °C for 1 hr to allow excess wax to melt before leaving to cool at room temperature. Sections were then deparaffinised in xylene and hydrated through a graded ethanol

series before being washed in PBS. Sections were then post-fixed in 4 % PFA/PBS before washing with buffer containing Proteinase K (PK; Qiagen UK Ltd, Manchester, UK) (1 µg/ml PK, 50 mM Tris-HCl [pH 7.5]; 6.25 mM EDTA in autoclaved water; Thermo Fisher Scientific, Oxford, UK) at 37 °C for 12 min. Sections were then post-fixed in 4 % PFA/PBS before washing with PBS/Tween 0.1 % and then acetylated (1.325 % triethanolamine [pH 7–8]; 0.25 % acetic anhydride; 0.175 % HCl in autoclaved water; VWR Chemicals, Lutterworth, UK) for 10 min. Sections were washed in PBS/0.1 % Tween-20 and in 5 × SSC. Then sections were dried, placed in a humidified chamber, and covered with hybridisation buffer (50 % formamide; 5 × SSC; 500 µg/ml yeast total RNA; 50 µg/ml heparin; 0.1 % Tween-20 in autoclaved water) at room temperature for 2 hr. ArSK/CCK precursor sense and antisense probes (500–1000 ng/ml) were denatured in hybridisation buffer at 80 °C and placed on ice before adding remaining hybridisation buffer and applying 100 µl probe solution per slide. Slides were covered with a piece of Parafilm (Bemis, Terre Haute, IN) and then placed in a humidified chamber at 65 °C overnight. Sections were then washed in 0.2 × SSC at 65 °C, in 0.2 × SSC at room temperature and equilibrated in buffer B1 (10 mM Tris-HCl [pH 7.5]; 150 mM NaCl in autoclaved water). Sections were covered in buffer B1/5 % goat serum and placed in a humidified chamber at room temperature for 2 hr. Sections were then dried and covered in an alkaline phosphatase-conjugated anti-DIG antibody (1:3000; Roche, Mannheim, Germany) in buffer B1/2.5 % goat serum at 4 °C overnight. Slides were washed in buffer B1 and then equilibrated in buffer B3 (100 mM Tris-HCl [pH 9.5]; 100 mM NaCl; 50 mM MgCl2 in autoclaved water). Sections were then covered in buffer B3/0.1 % Tween-20 with nitro-blue tetrazolium chloride (Sigma-Aldrich, Gillingham, UK) (75 mg/ml in 70 % dimethylformamide) and 5-bromo-4-chloro-3'-indolylphosphate-p-toluidine salt (BCIP; Sigma-Aldrich, Gillingham, UK) substrate solution (50 mg/ml BCIP in autoclaved water) until strong staining was observed. The slides were washed in distilled water to stop the staining reaction and then were dried on a hot plate before rinsing in 100 % ethanol and Histo-Clear (National Diagnostics, Fisher Scientific UK Ltd, Loughborough, UK). Sections were mounted with a coverslip on HistoMount solution (National Diagnostics, Fisher Scientific UK Ltd, Loughborough, UK) for long-term storage.

## Mass spectrometry

Extracts of *A. rubens* radial nerve cords were prepared and analysed using mass spectrometry (NanoLC-ESI-MS/MS), as described in detail previously for the *A. rubens* relaxin-like gonad stimulating peptide, which contains disulphide bridges (*Lin et al., 2017b*). Aliquots of radial nerve cord extract were not treated with trypsin but were subjected to reduction to break disulphide bridges (using 100 mM dithiothreitol; Sigma-Aldrich, Gillingham, UK) followed by alkylation of cysteine residues (using 200 mM iodoacetamide; Sigma-Aldrich, Gillingham, UK). Raw data were converted to Mascot generic format using MSConvert in ProteoWizard Toolkit (version 3.0.5759) (*Kessner et al., 2008*). MS spectra were searched with Mascot engine (Matrix Science, version 2.4.1) (*Nesvizhskii et al., 2003*) against a database comprising 40 *A. rubens* neuropeptide precursor proteins, including ArSK/CCKP (*Semmens et al., 2016*), all proteins in GenBank from species belonging to the family Asteriidae and the common Repository of Adventitious Proteins Database (http://www.thegpm.org/cRAP/index. html). A no-enzyme search was performed with up to two missed cleavages and carbamidomethyl as a fixed modification. Post-translational amidation of C-terminal glycine residues, pyroglutamylation of N-terminal glutamine residues, sulphation of tyrosine residues, and oxidation were included as variable modifications. Precursor mass tolerance was 10 ppm and product ions were searched at 0.8 Da tolerances.

Scaffold (RRID:SCR_014345; version Scaffold_4.6.1, Proteome Software Inc) was used to validate MS/MS-based peptide and protein identifications. Peptide identifications were accepted if they could be established at greater than 95.0 % probability by the scaffold local FDR algorithm. Protein identifications were accepted if they could be established at greater than 95.0 % probability and contained at least two identified peptides.

## Alignment of the *A. rubens* SK/CCK-type neuropeptides ArSK/CCK1 and ArSK/CCK2 with SK/CCK-type peptides from other taxa

Having used mass spectrometry to confirm the structures of the mature peptides ArSK/CCK1 and ArSK/CCK2 that are derived from ArSK/CCKP, the sequences of ArSK/CCK1 and ArSK/CCK2 were aligned with SK/CCK-type peptides from other taxa to investigate the occurrence of evolutionarily conserved

residues. The alignment was generated using MAFFT (RRID:SCR_011811; version 7) with the following parameters (BLOSUM62, 200 PAM/K = 2) and highlighted using BOXSHADE (http://www.ch.embnet.org/software/BOX_form.html) using 70 % conservation as a minimum for highlighting. The accession numbers for the sequences used for this analysis are shown in *Figure 1—source data 1*.

## Identification of a transcript encoding an *A. rubens* SK/CCK-type receptor

A transcript encoding an *A. rubens* SK/CCK-type receptor (ArSK/CCKR) was identified by tBLASTn analysis of the *A. rubens* radial nerve cord transcriptome data (*Semmens et al., 2016*), using SequenceServer (https://www.sequenceserver.com; *Priyam et al., 2015*) and a *Strongylocentrotus purpuratus* SK/CCK-type receptor (accession number XP_782630.3) as the query sequence. To investigate in more detail the relationship of ArSK/CCKR with SK/CCK-type receptors that have been identified in other taxa, phylogenetic analyses were performed using the maximum likelihood method. The sequences of ArSK/CCKR and SK/CCK-type receptors from a variety of taxa were aligned using MUSCLE (RRID:SCR_011812; iterative, 10 iterations, UPGMB as clustering method) (*Edgar, 2004*). The maximum likelihood tree was generated using IQ-tree web server (1000 bootstrap replicates, LG + F + I + G4 substitution model; *Trifinopoulos et al., 2016*). The accession numbers of the protein sequences that were used for this analysis are listed in *Figure 2—source data 1*.

## Cell lines and pharmacological characterization of ArSK/CCKR

To enable testing of ArSK/CCK1 and ArSK/CCK2 as candidate ligands for ArSK/CCKR, a full-length cDNA encoding ArSK/CCKR was synthesized by GenScript (Piscataway, NJ) and cloned into pcDNA 3.1+ vector (Invitrogen, Thermo Fisher Scientific, Waltham, MA). A partial Kozak translation initiation sequence (CACC) was introduced upstream to the start codon (ATG). CHO-K1 cells stably expressing the mitochondrial targeted calcium-sensitive bioluminescent reporter GFP-aequorin fusion protein (G5A) (*Baubet et al., 2000*) were used as a heterologous expression system for ArSK/CCKR. These cells have been used previously for neuropeptide receptor deorphanisation (*Bauknecht and Jékely, 2015*) and were generously supplied to us by Dr Gáspár Jékely (University of Exeter). The cell line was generated using the CHO-K1 cell line from Sigma-Aldrich (85051005; RRID: CVCL_0214), which is certified by the European Collection of Authenticated Cell Cultures (ECACC), and expression of the GFP-aequorin fusion protein (G5A) in the cell line was determined by functionality in displaying calcium-activated luminescence. Using the MycoAlert Mycoplasma Detection Kit (Lonza; LT07-118) cells were tested and found not to be contaminated with mycoplasma. Cells were cultured, co-transfected with the pcDNA 3.1+ vector containing the ArSK/CCKR cDNA sequence and plasmids encoding the promiscuous human G-protein $G_{\alpha}16$. Then bioluminescence-based receptor assays were performed, as described previously for *A. rubens* luqin-type receptors (*Yañez-Guerra et al., 2018*).

SK/CCK-type peptides in other taxa have a sulphated tyrosine residue that is important for bioactivity and therefore the ArSK/CCK1 and ArSK/CCK2 peptides were synthesized (Peptide Protein Research Ltd, Fareham, UK) with sulphated tyrosine residues: pQSKVDDY(SO$_3$H)GHGLFW-NH$_2$ (ArSK/CCK1), and GGDDQY(SO$_3$H)GFGLFF-NH$_2$ (ArSK/CCK2). Furthermore, to assess the requirement of tyrosine sulphation for receptor activation and bioactivity, a non-sulphated form of ArSK/CCK2 was also synthesized: GGDDQYGFGLFF-NH$_2$ [ArSK/CCK2(ns)]. The peptides were diluted in distilled water and tested as candidate ligands for ArSK/CCKR at concentrations ranging from $3 \times 10^{-17}$ to $10^{-4}$ M. Concentration-response data were determined as a percentage of the highest response for each peptide (100 % activation). EC$_{50}$ values were calculated from concentration-response curves based on four to six independent transfections and averaging two to three replicates in each transfection using Prism 6.0 (RRID:SCR_002798; GraphPad software, La Jolla, CA). Cells transfected with an empty vector were used for control experiments. Other *A. rubens* neuropeptides (Luqin: EEKTRFPKFMRW-NH$_2$ (ArLQ); tachykinin-like peptide 2: GGGVPHVFQSGGIFG-NH$_2$ (ArTK2); *Semmens et al., 2016*; *Yañez-Guerra et al., 2018*) were tested at a concentration of 10 μM to assess the specificity of receptor activation.

## Generation and characterisation of antibodies to ArSK/CCK1 and ArSK/CCKR

To generate antibodies to ArSK/CCK1, an N-terminally truncated peptide analog of ArSK/CCK1 with the addition of a reactive N-terminal lysine residue was synthesized (KY(SO$_3$H)GHGLFW-NH$_2$ , Peptide Protein Research Ltd, Fareham, UK). To generate antibodies to ArSK/CCKR, a peptide corresponding to the C-terminal 15 amino acids of ArSK/CCK-R with addition of an N-terminal reactive lysine residue was synthesized (KPSPTNYTNVSSDSSV, Peptide Protein Research Ltd, Fareham, UK). These peptides were conjugated to porcine thyroglobulin (Sigma-Aldrich, Gillingham, UK) as a carrier protein using 5 % glutaraldehyde (Sigma-Aldrich, Gillingham, UK) in phosphate buffer (0.1 M; pH 7.2) and the conjugate was used for immunisation of a rabbit for ArSK/CCK1 (70 -day protocol; Charles River Biologics, Romans, France) and a guinea pig for ArSK/CCKR (56 -day protocol; Charles River Biologics, Romans, France). The antigens were emulsified in Freund's complete adjuvant for primary immunisations (~100 nmol antigen peptide) and in Freund's incomplete adjuvant for three booster immunisations (~50 nmol antigen peptide). The presence of antibodies to the antigen peptides in post-immunisation serum samples was assessed using an enzyme-linked immunosorbent assay (ELISA), in comparison with pre-immune serum (*Figure 5—figure supplement 1a*; *Figure 6—figure supplement 1*). Antibodies to the antigen peptides were purified from the final bleed antiserum by affinity purification using the AminoLink Plus Immobilization Kit (Thermo Fisher Scientific, Waltham, MA), with bound antibodies eluted using glycine elution buffer (6.3 ml of 100 mM glycine [VWR Chemicals, Leicestershire, UK] and 0.7 ml of Tris [1 M, pH = 7.0]) and trimethylamine (TEA) elution buffer (6.3 ml of TEA [Sigma-Aldrich, Gillingham, UK] and 0.7 ml of Tris [1 M, pH = 7.0]). Eluates were dialyzed and sodium azide (0.1%) was added for long-term storage of the affinity-purified antibodies at 4 °C. The specificity of ArSK/CCK1 antibodies eluted with TEA, which were subsequently used for immunohistochemistry (see below), was assessed by ELISA by testing them at a concentration of 1:10 with the following synthetic peptides (100 µl at a concentration of 1 µM dissolved in carbonate/bicarbonate buffer [25 mM sodium carbonate, 25 mM sodium bicarbonate, pH = 9.8]): ArSK/CCK1, ArSK/CCK2, ArSK/CCK2(ns) and the *A. rubens* luqin-type peptide ArLQ (*Figure 5—figure supplement 1b*). The rabbit antiserum to ArSK/CCK1 has been assigned the RRID:AB_2877176 and the guinea pig antiserum to ArSK/CCKR has been assigned the RRID:AB_2891354.

## Immunohistochemical localisation of ArSK/CCK1 in *A. rubens*

Small specimens of *A. rubens* (<6 cm diameter) were fixed by immersion in seawater Bouin's fluid (75 % saturated picric acid [Sigma-Aldrich, Gillingham, UK] in seawater, 25 % formaldehyde, 5 % acetic acid) for 3–4 days at 4°C and then were decalcified for a week using a 2 % ascorbic acid/0.3 M sodium chloride solution, dehydrated and embedded in paraffin wax. Sections of the arms and the central disk region (8 µm; transverse or horizontal) were cut using a microtome (RM 2145, Leica Microsystems [UK], Milton Keynes, UK) and mounted on chrome alum/gelatin-coated microscope slides. Paraffin wax was removed by immersion of slides in xylene, and then slides were immersed in 100 % ethanol. Endogenous peroxidase activity was quenched using a 0.3 % hydrogen peroxide (VWT Chemicals, Leicestershire, UK)/methanol solution for 30 min. Subsequently, the slides were rehydrated through a graded ethanol series (90%, 70%, and 50%) and distilled water, blocked in 5 % goat serum (NGS; Sigma-Aldrich, Gillingham, UK) made up in PBS containing 0.1 % Tween (PBST). Then, the slides were incubated overnight with affinity-purified rabbit antibodies to ArSK/CCK1 (TEA fraction diluted 1:10 in 5 % NGS/PBST). Following a series of washes in PBST, indirect immunohistochemical detection was carried out using Peroxidase-AffiniPure Goat Anti-Rabbit IgG (H + L) conjugated to Horseradish Peroxidase (RRID: AB_2313567; Jackson ImmunoResearch, West Grove, PA) diluted 1:1000 in 2 % NGS/PBST. Bound antibodies were revealed using a solution containing 0.015 % hydrogen peroxide, 0.05 % diaminobenzidine (VWR Chemicals, Leicestershire, UK) and 0.05 % nickel chloride (Sigma-Aldrich, Gillingham, UK) in PBS. When strong staining was observed, sections were washed in distilled water, dehydrated through a graded ethanol series (50%, 70%, 90%, and 100%), and washed in xylene before being mounted with coverslips on DPX mounting medium (Thermo Fisher Scientific, Waltham, MA). Immunostaining was not observed in negative control tests without the primary antibodies or with primary antibodies that had been pre-adsorbed with the antigen peptide at a concentration of 20 µM (data not shown).

## Localisation of ArSK/CCK1 and ArSK/CCKR in *A. rubens* using double immunofluorescence labelling

To visualise and compare expression of ArSK/CCK1 and ArSK/CCKR in *A. rubens*, sections of arms and the central disk region were prepared as described above and then were first incubated with affinity-purified rabbit antibodies to ArSK/CCK1 (TEA fraction diluted 1:15 in 5 % NGS/PBST) overnight at 4 °C. Then, following washes in PBST, the sections were incubated with affinity-purified guinea pig antibodies to ArSK/CCKR (TEA fraction diluted 1:3.3 in 5 % NGS/PBST) for 3–5 days at 4 °C. Then, following washes in PBST, the sections were incubated for 3 hr with Alexa Fluor 488-conjugated AffiniPure Goat Anti-Rabbit IgG (H + L) (RRID: AB_2338046; Jackson ImmunoResearch, West Grove, PA) and Cy3-conjugated AffiniPure Goat Anti-Guinea Pig IgG (H + L) (RRID: AB_2337423; Jackson ImmunoResearch, West Grove, PA), which were diluted 1:500 and 1:300 respectively in 2 % normal goat serum in PBST. Following washes in PBST, the slides were mounted with coverslips using Abcam Fluroshield Mounting Medium with DAPI (Thermo Fisher Scientific, Waltham, MA).

## Imaging

Photographs of sections processed for mRNA in situ hybridization or peroxidase-based immunohistochemistry were captured using a QIClick CCD Color Camera (Qimaging, British Columbia, CA) linked to a DMRA2 light microscope (Leica), utilising Volocity version 6.3.1 image analysis software (PerkinElmer, Seer Green, UK) running on iMac computer (27-inch, Late 2013 model with OS X Yosemite, version 10.10). To capture images of sections labelled using fluorescence immunohistochemistry, a Leica SP5 confocal microscope was used in combination with the Leica Application Suite Advanced Fluorescence (LAS AF; version 2.6.0.7266) programme. The settings for capturing images were as follows: image format, 1024 × 1024; scan speed, 200 Hz; frame average, 6; and line average accumulation, 3. Images were acquired using sequential scanning and then merged, with contrast and level adjustment, using ImageJ (RRID:SCR_003070). Representative photographs, based on analysis of sections from at least three animals for each technique, were assembled as montages using Adobe Photoshop CC (RRID:SCR_014199; version 19.1.4, ×64) running on a MacBook Pro computer (13-inch, early 2015 model with OS Mojave version 10.14.3).

## In vitro pharmacology

Informed by analysis of the expression of ArSK/CCKP transcripts, ArSK/CCK1 and ArSK/CCKR in *A. rubens*, both ArSK/CCK1 and ArSK/CCK2 were tested for myoactivity on cardiac stomach, tube foot, and apical muscle preparations dissected from specimens of *A. rubens* (n = 5–9, 8–10, and 20–23, respectively) and set up in a 20 ml organ bath, as described previously (*Elphick et al., 1995*; *Melarange and Elphick, 2003*; *Tian et al., 2017*). Effects of peptides on preparations were tested and recorded using an isotonic transducer (MLT0015, ADInstruments Pty Ltd) connected to a bridge amplifier (FE221 Bridge Amp, ADInstruments Pty Ltd) linked to PowerLab data acquisition hardware (2/36, ADInstruments Pty Ltd). Data were collected and analysed using LabChart (RRID:SCR_017551; v8.0.7) software installed on a laptop computer (Lenovo E540, Windows 7 Professional). Stock solutions of synthetic peptides were prepared in distilled water and added to the organ bath to achieve final concentrations ranging from 0.1 nM to 1 μM. To assess the viability of preparations and to enable normalization of responses to ArSK/CCK1 or ArSK/CCK2, the starfish neuropeptide NGFFYamide (100 nM) was tested on cardiac stomach preparations and acetylcholine (10 μM) was tested on tube foot and apical muscle preparations. To assess the importance of tyrosine sulphation for peptide bioactivity, a non-sulphated analog of ArSK/CCK2 (ArSK/CCK2(ns)) was also tested on cardiac stomach (n = 5), tube foot, and apical muscle preparations (data not shown).

## In vivo pharmacology: testing ArSK/CCK1 and ArSK/CCK2 as cardiac stomach retractants

In vitro pharmacological experiments revealed that both ArSK/CCK1 and ArSK/CCK2 cause contraction of cardiac stomach preparations. Previous studies have revealed that the neuropeptide NGFFYamide causes cardiac stomach contraction in vitro and also triggers retraction of the everted cardiac stomach when injected into *A. rubens* in vivo (*Semmens et al., 2013*). Therefore, experiments were performed to investigate if ArSK/CCK1 and ArSK/CCK2 also trigger cardiac stomach retraction in *A. rubens*. Twenty specimens of *A. rubens*, which had been withheld from a food supply for 1 week, were placed

in a glass tank containing 2 % magnesium chloride (MgCl$_2$; Sigma-Aldrich, Gillingham, UK) dissolved in seawater, which acts as a muscle relaxant in marine invertebrates (*Mayer, 1909*). This treatment conveniently and reproducibly causes eversion of the cardiac stomach in *A. rubens*, typically within a period of ~30 min (*Semmens et al., 2013*). Hamilton 75 N 10 µl syringes (Sigma-Aldrich, Gillingham, UK) were used to inject test compounds into the perivisceral coelom of animals, inserting the needle through the aboral body wall of the arms proximal to the junctions with the central disk region. Care was taken to inject neuropeptides (ArSK/CCK1, ArSK/CCK2 and ArSK/CCK2(ns)) or distilled water (control) into the perivisceral coelom and not into the cardiac stomach. All animals were first injected with 10 µl of distilled water (control) and video-recorded for 6 min. The same animals were then injected with 10 µl of 1 mM peptide (10 nmol) and video-recorded for 6 min. Static images from video recordings were captured at 30 s intervals from the time of injection. Then the two-dimensional area of everted cardiac stomach was measured from the images using the ImageJ software (version 1.0; http://rsb.info.nih.gov/ij) and normalized as a percentage of the area of cardiac stomach everted at the time of injection (T0).

## In vivo pharmacology: testing effects of ArSK/CCK1 and ArSK/CCK2 on feeding behaviour

Previous studies have revealed that the starfish neuropeptide NGFFYamide inhibits the onset of feeding behaviour of *A. rubens* on a prey species – the mussel *Mytilus edulis* (*Tinoco et al., 2018*). Here, the same methods employed by *Tinoco et al., 2018* were used to investigate if ArSK/CCK1 and/or ArSK/CCK2 affect feeding behaviour in starfish. Sixty-two adult starfish (n = 24 for ArSK/CCK1; n = 38 for ArSK/CCK2) that met the following criteria were used: (i) all five arms were intact, (ii) exhibited a normal righting response (*Lawrence and Cowell, 1996*), and (iii) after 24 days of starvation, exhibited normal feeding behaviour on a mussel. Then, starfish were fasted for 24 days and transferred to and kept individually in Plexiglas aquaria, as described previously (*Tinoco et al., 2018*). After 3 days of acclimation (27 days of starvation at this point) and at 10 a.m., these animals were then divided into a control group (to be injected with distilled water), with 13 animals used for the ArSK/CCK1 experiment (mean diameter of 12.4 ± 0.3 cm) and 19 animals used for the ArSK/CCK2 experiment (mean diameter of 12.9 ± 0.4 cm), and a test group (to be injected with ArSK/CCK1 or ArSK/CCK2), with 11 animals used for the ArSK/CCK1 experiment (mean diameter of 12.6 ± 0.3 cm) and 19 animals used for the ArSK/CCK2 experiment (mean diameter of 12.9 ± 0.5 cm). The starfish were then injected with 10 µl of distilled water (control group) or 10 µl of 1 mM ArSK/CCK1 or ArSK/CCK2 peptides (test group) to achieve an estimated final concentration in the perivisceral coelom of ~1 µM, which is the concentration at which ArSK/CCK peptides were found to have a maximal effect when tested on in vitro preparations of the cardiac stomach. The time taken for starfish to make first contact with a mussel (tube feet touching the mussel or time to touch the mussel), the number of attempts to touch, as well as the time to enclose the mussel (indicated by a feeding posture) were recorded. Starfish that were feeding after 24 hr were included in data analysis and any starfish in the control or test group that had not fed on a mussel after 24 hr were discarded from data analysis.

## Statistical analyses

Data were presented as means ± standard error of the mean (s.e.m.). The in vitro or in vivo pharmacological effects of starfish SK/CCK-type peptides on cardiac stomach, apical muscle, and tube foot preparations were analysed by two-way ANOVA, using type of substance tested and concentration/time as independent factors and Bonferroni's multiple comparison test. Apical muscle and tube foot data were transformed to logarithms to obtain a normal distribution and homogeneity of variances. The in vitro effects of ArSK/CCK1 and ArSK/CCK2 (1 µM) on cardiac stomach preparations were compared with the in vitro effect of NGFFYamide (100 nM) using a two-tailed Student's t-test. The effect of ArSK/CCK1 on feeding behaviour was analysed by two-tailed Mann-Whitney U-test (time to touch and time to enclose) because these data did not follow a normal distribution when analysed using the D'Agostino and Pearson omnibus normality test. The effect of ArSK/CCK2 on feeding behaviour was analysed by two-tailed Student's t-test (time to touch) or Welch's unequal variances t-test (time to enclose). Fisher's exact test was used to analyse the percentage of successful feeding after the first touch for control and treated starfish. Statistical analyses were carried out using Prism

6.0 (GraphPad software, La Jolla, CA) and differences were considered statistically significant at $p < 0.05$.

## Acknowledgements

This work was supported by grants awarded to MRE by BBSRC (BB/M001644/1) and the Leverhulme Trust (RPG-2018–200) and to AMJ by BBSRC (BB/M001032/1). LAYG was supported by a PhD studentship awarded by the Mexican Council of Science and Technology (CONACyT studentship no. 518612) and Queen Mary University of London. YZ was supported by a PhD studentship awarded by the China Scholarship Council and Queen Mary University of London. JD is currently a postdoctoral researcher supported by Fund for Scientific Research of Belgium (FRS-FNRS).

## Additional information

### Funding

| Funder | Grant reference number | Author |
|---|---|---|
| Biotechnology and Biological Sciences Research Council | BB/M001644/1 | Maurice R Elphick |
| Biotechnology and Biological Sciences Research Council | BB/M001032/1 | Alexandra M Jones |
| Leverhulme Trust | RPG-2018-200 | Maurice R Elphick |
| Consejo Nacional de Ciencia y Tecnología | 518612 | Luis Alfonso Yañez Guerra |
| China Scholarship Council | | Ya Zhang |
| Fund for Scientific Research - FNRS | | Jérôme Delroisse |
| Queen Mary University of London | | Luis Alfonso Yañez Guerra Ya Zhang |

The funders had no role in study design, data collection and interpretation, or the decision to submit the work for publication.

### Author contributions

Ana B Tinoco, Conceptualization, Data curation, Formal analysis, Investigation, Methodology, Visualization, Writing – original draft, Writing – review and editing; Antón Barreiro-Iglesias, Luis Alfonso Yañez Guerra, Jérôme Delroisse, Conceptualization, Data curation, Formal analysis, Investigation, Methodology, Visualization, Writing – review and editing; Ya Zhang, Elizabeth F Gunner, Investigation, Methodology; Cleidiane G Zampronio, Investigation, Methodology, Visualization, Writing – review and editing; Alexandra M Jones, Funding acquisition, Project administration, Supervision, Writing – review and editing; Michaela Egertová, Investigation, Methodology, Visualization; Maurice R Elphick, Conceptualization, Data curation, Funding acquisition, Methodology, Project administration, Supervision, Writing – original draft, Writing – review and editing

### Author ORCIDs

Ana B Tinoco (iD) http://orcid.org/0000-0003-0525-4475
Antón Barreiro-Iglesias (iD) http://orcid.org/0000-0002-7507-080X
Luis Alfonso Yañez Guerra (iD) http://orcid.org/0000-0002-2523-1310
Jérôme Delroisse (iD) http://orcid.org/0000-0002-9233-6470
Ya Zhang (iD) http://orcid.org/0000-0002-2158-0660
Cleidiane G Zampronio (iD) http://orcid.org/0000-0003-0934-0792
Alexandra M Jones (iD) http://orcid.org/0000-0003-2571-8708
Maurice R Elphick (iD) http://orcid.org/0000-0002-9169-0048

Decision letter and Author response
Decision letter https://doi.org/10.7554/eLife.65667.sa1
Author response https://doi.org/10.7554/eLife.65667.sa2

## Additional files

### Supplementary files
• Transparent reporting form

### Data availability
The sequences of cDNAs encoding ArSK/CCKP and ArSK/CCKR have been deposited in GenBank under accession numbers, KT601716 and MW261740, respectively. All data generated or analysed during this study are included in the manuscript and supporting files. Source data files have been provided for Figures 1,2,3,7,8 and 9.

The following dataset was generated:

| Author(s) | Year | Dataset title | Dataset URL | Database and Identifier |
|---|---|---|---|---|
| Semmens DC, Mirabeau O, Moghul I, Pancholi MR, Wurm Y, Elphick MR | 2016 | Asterias rubens cholecystokinin-type precursor, mRNA, complete cds | https://www.ncbi.nlm.nih.gov/nuccore/KT601716 | NCBI GenBank, KT601716 |
| Tinoco AB, Barreiro-Iglesias A, Yañez Guerra LA, Delroisse J, Zhang Y, Gunner EF, Zampronio CG, Jones AM, Egertova M, Elphick MR | 2021 | Asterias rubens cholecystokinin-type receptor (CCKR) mRNA, complete cds | https://www.ncbi.nlm.nih.gov/nuccore/MW261740 | NCBI GenBank, MW261740 |

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
