## [Decision Letter]

**Acceptance summary:**

The study provides new insight into the evolution and comparative physiology of SK/CCK-type neuropeptide signalling in Bilateria. It is thus of interest to neuroscientists, endocrinologists and evolutionary biologists. More specifically, the paper covers detailed localization analyses based on the prepropeptide transcript, as well as the immunohistochemistry of Aru-SK/CCK1 and Aru-SK/CCKR receptor. It furthermore includes Mass Spec verification of the peptides, receptor deorphanization and functional tests that reveal an involvement in the regulation of feeding behaviors in the starfish Asterias rubens.

**Decision letter after peer review:**

Thank you for submitting your article "Ancient role of cholecystokinin-type signaling in inhibitory regulation of feeding processes revealed in an echinoderm" for consideration by *eLife*. Your article has been reviewed by 3 peer reviewers, and the evaluation has been overseen by a Reviewing Editor and K VijayRaghavan as the Senior Editor. The following individual involved in the review of your submission has agreed to reveal their identity: Simon Webster (Reviewer #2).

The reviewers have discussed their reviews with one another, and the Reviewing Editor has drafted this to help you prepare a revised submission. Major concerns to be addressed in a revised version of the manuscript:

1. Given the rather general effects of Arub-CCKs on muscle contractions, data showing the spatial and temporal expression profiles of Arub-CCKR in target tissues would be highly beneficial to interpret the presented physiological data.

If the authors have meanwhile acquired a functioning antibody against the receptor, we would ask them to add such data. If this is not the case then whole-mount in situ hybridization and qPCR would already add very valuable insight. We do understand that there may be technical difficulties for whole-mount in situs in this case, in which case, do see if qRTPCR is feasible.

2. In Figure 3, the authors used CHO-K1 cells expressing aequorin as an assay system to determine the Arub-CCK1 and Arub-CCK-induced ca^2+^ response. What can the authors state about the ligand specificity/selectivity of Arub-CCKR in comparison with other neuropeptides from the starfish Asterias rubens and other cholecystokinin-type neuropeptides from different species?

3. There are substantial questions if naming the system "CCKs" is correct.

Mammals have two structurally related peptide hormones, cholecystokinin and gastrin, which can be further processed to smaller peptides of variable lengths. For these peptides to be active they need to have their C-terminal sequence GWMDFamide intact, which is, therefore, the hallmark of these gastrin/CCK peptide hormones. Also other chordates like amphibians and the tunicate *Ciona intestinalis* produce peptides (caerulein, respectively cionin) having the C-terminal GWMDFamide sequence. In addition to this C-terminal GWMDFamide hallmark, all chordate neuropeptides have a sulfated tyrosin residue located either one or two residues before the GWMDFamide sequence. Surprisingly, the GWMDFamide hallmark is absent in the two starfish peptides (none of these residues are there at corresponding positions), although a sulfated tyrosine is present at the correct position. The same situation is found in protostome invertebrates, such as insects, mollusks, and annelids. In these protostomes, however, the peptides are correctly named sulfakinins, due to a sulfated tyrosine residue located seven residues away from the C-terminus. Because the starfish peptides lack the GWMDFamide hallmark and, thus, much more resemble the protostome sulfakinins than mammalian CCKs, there is no justification to name them CCK's in the manuscrpt. Instead, they should be called sulfakinins. Sulfakinin is also a more appropriate name, because it refers to a structure, while CCK refers to a physiological action, namely the emptying of the gall bladder, which, of course, is not relevant in a starfish. Calling the starfish peptides CCK's is also leading to confusions, because it insinuates that the peptides would act like CCK and this is what is actually happening in the title of the manuscript: "Ancient role of cholecystokinin-type signalling in inhibitory regulation of feeding processes revealed in an echinoderm". Figures 6 and 7 of the manuscript show that living starfish contract their stomach after injection of the Arub -CCK's. However, also isolated muscle preparations that don't have any relation to feeding, such as tube foot preparations do contract after the addition of Arub -CCKs (Figure 6). This tells the reader of the manuscript that the Arub -CCK's are myoexcitatory, or in other words that they are"kinins", just like numerous other invertebrate and vertebrate neuropeptides are kinins. However, in the Discussion (line 824) the authors of this manuscript argue that the Arub -CCK's act like a human CCK. Their reasoning: Arub -CCK's in starfish fulfill the role of mammalian CCK, because both peptides are myoexitatory. But these arguments are, of course, not acceptable. In mammals , substance P, neuromedin, oxytocin, endothelin, acetylcholine, prostaglandin and many other neurohormones and transmitters activate smooth muscles. In insects about half of all isolated neuropeptides stimulate muscle contraction. Myoexcitation is a very unspecific property of neuropeptides and cannot be used as a criterium to establish a functional or evolutionary link or an evolutionarily conserved property between peptides. The conclusion that Arub-CCKs are the functional equivalents of human CCK, therefore, is not supported by the experiments from Figures 6 and 7. These experiments only show that the Arub-CCKs are kinins, which stimulate smooth muscle cells and inhibit feeding.

Therefore the following points are requested:

– re-name these peptides sulfakinins (Arub-SKs) or at least Arub-SKs/CCKs.

– Both in the Title and in the Discussion, the authors should omit suggesting a functional identity between Arub-CCKs and human CCK, because this leads to misunderstandings.

4. Resolutions of Figure 4 and 5. Make improved Figures 4 and 5, with a higher resolution. Alternatively, for Figure 5, the authors may consider improving their staining technique, for example by considering other fixation/permeabilization methods and by applying fluorescent secondary antibodies.

As the RNA localization also serves as a control that the antibody against Arub-CCK1 is likely specific, a more direct comparative juxtaposition of the whole-mount in situ images vs. antibody staining should be considered.

5. Please justify why a lot of gaps are introduced in the alignment of Figure 1. Gaps are often introduced to suggest (often in a biased way) structural similarities. For peptide ligands of GPCRs, the length of the peptides or the distance between two residues are often of crucial importance for binding to the receptor. The authors should also consider to provide an alignment with no gaps, which gives a better impression of structural relationships.

6. The Discussion is very verbose and unnecessarily narrative. Reduce it by 33% (from 9 to 6 manuscript pages) to make it more readable. For instance, the authors write that the DYG motiv seen in the Arub-CCKs is occurring in many other CCK's from other animal taxa (line 669). However, it is not present in the canonical human CCK's, gastrins, frog caerulein, and tunicate cionin. Therefore, omit this part of the Discussion.

Below are the individual reviews for your information only. There is no need to address them individually beyond what has been stated in our consensus summary above.

*Reviewer #1:*

Tinoco and colleagues describe the characterization of the cholecystokinin-type signaling system in the starfish Asterias rubens. Using bioinformatic and LC-MS-MS analysis, two neuropeptides, ArCCK1 and ArCCK2, derived from the precursor protein ArCCKP, were identified, and further experiments showed their expression in the nervous system, digestive system, tube feet and body wall. In addition, in a heterologous expression system of CHO-K1 cells, a CCK/SK-type receptor, ArCCKR, was identified as a cognate receptor for the sulphated neuropeptides, ArCCK1 and ArCCK2. Moreover, in vitro and in vivo bioassays demonstrated the biological activities of ArCCK1 and ArCCK2 to induce dose-dependent contraction of cardiac stomach, tube foot and body wall apical muscle and inhibit feeding behaviour in A. rubens.

Strengths:

1. There is the potential that identification of a CCK/SK-type receptor for the sulphated forms of ArCCK1 and ArCCK2 from the starfish Asterias rubens may be a step forward in understanding the novel signaling mechanisms and the physiological functions of cholecystokinin-type signaling.

2. The data derived from the in vitro and in vivo bioassays showing significant effects of neuropeptides, ArCCK1 and ArCCK2, on contraction of cardiac stomach, tube foot and body wall apical muscle and feeding behaviour in A. rubens represent the novel aspect of this manuscript, providing insights into the evolution and comparative physiology of several neuropeptide signalling systems.

Weaknesses:

1. The same CCK/SK-type neuropeptide signalling system has been well characterized in the nematode *C. elegans* (Ann. N.Y. Acad. Sci. 2009, 1163: 428-432), the mollusc C. gigas (Sci Rep. 2018, 8:16424) and the ascidian C. intestinalis (J Endocrinol. 2012, 213, 99-106), this makes the novelty of this work uncertain.

2. Figure 3: the data showed that the sulphated ArCCK1 and ArCCK2 induced ca^2+^ mobilization in CHO-K1 cells co-expressing mitochondrial targeted apoaequorin (G5A) and the promiscuous human G-protein Gα16 and ArCCKR, however, more data such as binding activity and ligand specificity are necessary to substantiate the conclusions.

3. ArCCKR has been identified as a cognate receptor for the sulphated ArCCK1 and ArCCK2, it would be good to show the G protein coupling and downstream signaling cascades.

4. To understand the physiological roles of the CCK-type signalling system in A. rubens, the data showing the spatial and temporal expression profiles of ArCCKR in target tissues are more important than that of neuropeptides, ArCCK1 and ArCCK2.

5. Figure 6, 7 and 8: the in vitro and in vivo bioassays showed effects of neuropeptides, ArCCK1 and ArCCK2 on the contraction of cardiac stomach, tube foot and apical muscle, cardiac stomach retraction and feeding behavior in A. rubens, however, there is no data to show that these effects are mediated by ArCCKR.

6. In Figure 3, the authors used CHO-K1 cells expressing aequorin as an assay system to determine the ArCCK1 and ArCCK-induced ca^2+^ response. The authors should use this same assay system to check the ligand specificity/selectivity of ArCCKR in comparison with other neuropeptides from the starfish Asterias rubens and other cholecystokinin-type neuropeptides from different species.

7. To further confirm the direct interaction between ArCCKR and neuropeptides, ArCCK1 and ArCCK, the authors should perform fluorescent-labeled ligand-based binding assay.

8. Like the data shown in Figure 4 and 5, it will be better for authors to generate antibodies against ArCCKR, and to combine with qRT-PCR to investigate the distribution and expression of ArCCKR in different tissues and different development stages. These data of ArCCKR distribution and expression will substantiate the conclusions.

9. In Figure 6, 7 and 8, the authors used in vitro and in vivo bioassays to determine effects of neuropeptides ArCCK1 and ArCCK on contraction of cardiac stomach, tube foot and apical muscle, cardiac stomach retraction and feeding behavior, however, if the authors can use siRNA-based knockdown of ArCCKR or anti-ArCCKR antibodies as controls, it will more convincing.

*Reviewer #2:*

Tinoco and colleagues investigated the occurrence of cholecystokinin type (CCK) peptides and have functionally deorphaned their cognate receptor in an invertebrate deuterostome, the starfish, Asterias rubens. Of interest are the findings that both identified CCKs activate the identified CCK-type receptor, but only in their sulfated forms, a phenomenon that is also seen in the bioassays. The work further details the wide distribution of these peptides in the nervous system and body tissues of this organism by detailed in-situ hybridization and immunohistochemistry, techniques which are technically demanding in these animals, and often difficult to interpret, given the unusual morphology of these organisms. The morphology of CCK expressing neurons, strongly suggests further pleiotropic functions, that await discovery. Some beautifully compelling behavioral and pharmacologically-based results, have been presented in a novel way, which convincingly show that the very unusual feeding behaviors of starfish, whereby the stomach is everted, can be inhibited by these CCK-type peptides, at physiologically relevant levels and the onset of feeding can likewise be inhibited. These important results beautifully link analogous processes involved in feeding, digestion and satiety in vertebrates, thus highlighting conservation of structure with function across the Bilateria from the viewpoint of evolution of peptides and receptors. The data have been faultlessly collected, to an exceptional standard, and the work nicely integrates a variety of techniques from the molecule to whole organism- this type of comprehensive work is rarely performed or appreciated- and adds interest to a broad readership.

The paper has been faultlessly written in a comprehensive and engaging style: I very much enjoyed reading it. Whilst there are many references, I believe that these are all justified, particularly since the paper will be of broad interest to vertebrate and invertebrate neuroscientists and evolutionary biologists.

The statistical tests have been properly applied, and there is clear justification for the tests used, when, for example, criteria for normality have not been met.

I particularly liked the highly detailed and comprehensive Materials and methods section.

*Reviewer #3:*

This is an interesting paper that identifies two peptides (Arub-CCK-1 and -2; also called arCCK1 and -2 in the paper), and one associated G protein-coupled receptor (GPCR) in the starfish Asterias rubens that appear to play a major role in feeding in that animal. The GPCR is functionally expressed in mammalian (Chinese Hamster Ovary) cells and gets activated by nanomolar concentrations of the two peptides, thereby proving that the GPCR is indeed the intrinsic receptor for Arub-CCK-1 and -2. Based on the phylogenetic position of starfishes in the Animal Kingdom, this paper will be a valuable contribution to understanding the evolution of the neuropeptide transmitters and their receptors that control feeding.

The major strength of this paper is that it proves that the starfish GPCR (named by the authors ArCCKR) is indeed the receptor for the two peptides, Arub-CCK-1 and -2. Their dose-response curves in Figure 3 very clearly show this. Also, the phylogenetic tree given in Figure 2, nicely shows that the starfish receptor ArCCKR is an orthologue of the human receptors for cholecystokinin (CCKR1) and gastrin (CCKR2), which are classical and well-established human neuroendocrine peptide hormones involved in feeding and digestion. These data, therefore, strongly suggest that also the starfish receptor ArCCKR might be involved in feeding. This idea has subsequently been tested by physiological in vitro and in vivo studies on whole living starfish and various isolated starfish preparations, including the animal's stomach. The results (Figures 6 and 7) show that, after addition or injection of the peptides, starfishes stop eating (=retract their everted stomachs, which is a sign of feeding cessation) and stop approaching and attacking prey.

The manuscript, however, also has major weaknesses. Mammals have two structurally related peptide hormones, cholecystokinin and gastrin, which can be further processed to smaller peptides of variable lengths. For these peptides to be active they need to have their C-terminal sequence GWMDFamide intact, which is, therefore, the hallmark of these gastrin/CCK peptide hormones. Also other chordates like amphibians and the tunicate *Ciona intestinalis* produce peptides (caerulein, respectively cionin) having the C-terminal GWMDFamide sequence. In addition to this C-terminal GWMDFamide hallmark, all chordate neuropeptides have a sulfated tyrosin residue located either one or two residues before the GWMDFamide sequence. Surprisingly, the GWMDFamide hallmark is absent in the two starfish peptides (none of these residues are there at corresponding positions), although a sulfated tyrosine is present at the correct position. The same situation is found in protostome invertebrates, such as insects, mollusks, and annelids. In these protostomes, however, the peptides are correctly named sulfakinins, due to a sulfated tyrosine residue located seven residues away from the C-terminus. Because the starfish peptides lack the GWMDFamide hallmark and, thus, much more resemble the protostome sulfakinins than mammalian CCKs, there is no justification to name them CCK's in the manuscript. Instead, they should be called sulfakinins. Sulfakinin is also a more appropriate name, because it refers to a structure, while CCK refers to a physiological action, namely the emptying of the gall bladder, which, of course, is not relevant in a starfish. Calling the starfish peptides CCK's is also leading to confusions, because it insinuates that the peptides would act like CCK and this is what is actually happening in the title of the manuscript: "Ancient role of cholecystokinin-type signalling in inhibitory regulation of feeding processes revealed in an echinoderm". Figures 6 and 7 of the manuscript show that living starfish contract their stomach after injection of the arub-CCK's. However, also isolated muscle preparations that don't have any relation to feeding, such as tube foot preparations do contract after addition of arub-CCKs (Figure 6). As a reader of this manuscript, this tells me that the arub-CCK's are myoexcitatory, or in other words that they are"kinins", just like numerous other invertebrate and vertebrate neuropeptides are kinins. However, in the Discussion (line 824) the authors of this manuscript argue that the arub-CCK's act like a human CCK. Their reasoning: arub-CCK's in starfish fulfill the role of mammalian CCK, because both peptides are myoexitatory. But these arguments are, of course, not acceptable. In mammals , substance P, neuromedin, oxytocin, endothelin, acetylcholine, prostaglandin and many other neurohormones and transmitters activate smooth muscles. In insects about half of all isolated neuropeptides stimulate muscle contraction. Myoexcitation is a very unspecific property of neuropeptides and cannot be used as a criterium to establish a functional or evolutionary link or an evolutionarily conserved property between peptides. The conclusion that arub-CCKs are the functional equivalents of human CCK, therefore, is not supported by the experiments from Figures 6 and 7. These experiments only tell us, that the arub-CCKs are kinins, which stimulate smooth muscle cells and inhibit feeding.

Other weaknesses concern Figure 4 and Figure 5, which have very low (pixel) resolution and are, therefore, hard to interpret. In Figure 5, which should show the immunocytochemical staining of arub-CCK-1 and -2 in the nervous system of various parts of the starfish, it is difficult for me to recognize neurons. Here, the authors have used an inappropriate method or a low-affinity (inappropriate) antibody, as immunocytochemistry is a well-established technique, normally giving excellent staining results.

I have the following recommendations for improving the submitted manuscript:

1. Because the claim by the authors that the arub-CCK's are the functional equivalents of human CCK is not supported by their data (see Public Review), I recommend to name these peptides sulfakinins (arub-SK's).

2. Both in the Title and in the Discussion, the authors should omit suggesting a functional identity between arub-CCK's and human CCK, because this leads to confusion and misunderstandings.

3. The Discussion is very verbose and unnecessarily narrative. Reduce it by 33% (from 9 to 6 manuscript pages) to make it more readable and preventing the audience from falling asleep.

4. In the Discussion, the authors write that the DYG motiv seen in the arub-CCKs is occurring in many other CCK's from other animal taxa (line 669). However, it is not present in the canonical human CCK's, gastrins, frog caerulein, and tunicate cionin. Therefore, omit this part of the Discussion.

5. Make improved Figures 4 and 5, with a higher resolution. For Figure 5, use a better primary antibody and a better staining technique, for example by applying fluorescent secondary antibodies.

6. I don't like alignments as given in Figure 1, where a lot of gaps are introduced to suggest (often in a biased way) structural similarities. For peptide ligands of GPCRs, the length of the peptides or the distance between two residues are often of crucial importance for binding to the receptor. Therefore make an alignment with no gaps, which is free of bias and, therefore, more realistic. This gives a better impression of structural relationships.

[Editors' note: further revisions were suggested prior to acceptance, as described below.]

The manuscript has been improved significantly, however there is one remaining issue that we would ask you to address, as outlined below:

Reviewer 3 brought up the aspect of re-naming again (for detailed argumentation see their comment). In essence the request is to change the name from CCK/SKP to SK/CCK, as this request then also gained support in the discussion among reviewers/RE. We acknowledge that the name CCK had already been used in 2016, but think that the recognition won't be hampered by switching the order.

---

## [Author Response]

The reviewers have discussed their reviews with one another, and the Reviewing Editor has drafted this to help you prepare a revised submission. Major concerns to be addressed in a revised version of the manuscript:1. Given the rather general effects of Arub-CCKs on muscle contractions, data showing the spatial and temporal expression profiles of Arub-CCKR in target tissues would be highly beneficial to interpret the presented physiological data.If the authors have meanwhile acquired a functioning antibody against the receptor, we would ask them to add such data. If this is not the case then whole-mount in situ hybridization and qPCR would already add very valuable insight. We do understand that there may be technical difficulties for whole-mount in situs in this case, in which case, do see if qRTPCR is feasible.

In response to this feedback, we have investigated the expression of ArCCKR in *A. rubens* (now referred to as ArCCK/SKR; see below) and compared its expression with ArCCK1 (now referred to as ArCCK/SK1; see below) using double immunofluorescence labelling. To accomplish this, antibodies to a C-terminal region of ArCCK/SKR generated in guinea pig (Figure 6 —figure supplement 1) were affinity-purified and used in combination with affinity-purified rabbit antibodies to ArCCK/SK1 (see lines 901-930 for details of methods). Importantly, this has revealed the expression pattern of ArCCK/SKR and its relationship to ArCCK/SK1 expression in the CNS (circumoral nerve ring and radial nerve cords) and in organs where ArCCK/SK1 exhibits myoactivity – tube feet, apical muscle and cardiac stomach. These new data are presented in Figure 6 and text describing and interpreting the functional significance of the data has been added to the results (lines 292-329) and discussion (lines 483-583).

2. In Figure 3, the authors used CHO-K1 cells expressing aequorin as an assay system to determine the Arub-CCK1 and Arub-CCK-induced ca^2+^ response. What can the authors state about the ligand specificity/selectivity of Arub-CCKR in comparison with other neuropeptides from the starfish Asterias rubens and other cholecystokinin-type neuropeptides from different species?

We agree that it is important to assess the ligand specificity/selectivity of ArCCK/SKR in starfish as a basis for assessment of the pharmacological actions of ArCCK/SK1 and ArCCK/SK2. Evidence of the specificity/selectivity of ArCCK/SKR for the sulphated peptides ArCCK/SK1 and ArCCK/SK2 is provided by our finding that the EC_50_ for the non-sulphated form ArCCK/SK2 [ArCCK/SK29(ns)] is 48 µM, a more than 100,000 fold reduction in potency compared to the sulphated peptides (see Figure 3). Furthermore, we also tested two *A. rubens* neuropeptides that share some C-terminal sequence similarity with ArCCK/SK1 and ArCCK/SK2 – the SALMFamide-type neuropeptide S2 (SGPYSFNSGLTF-NH_2_) and an *A. rubens* tachykinin-like peptide GxFamide2 (GGGVPHVFQSGGIF-NH_2_). Importantly, neither of these peptides caused activation of ArCCK/SKR, even at concentrations as high as 10 µM (Figure 3 —figure supplement 1), providing important evidence of the selectivity/specificity of ArCCK/SKR as a receptor for ArCCK/SK1 and ArCCK/SK2. Investigation if CCK/SK-type peptides from other taxa can act as ligands for ArCCK/SKR was not investigated in this study because of its lack of physiological relevance; however, it would be of interest for a future study that specifically investigates the structure-activity relationships of CCK/SK-type peptides as ligands for CCK/SK-type receptors from different taxa from a comparative/evolutionary perspective.

3. There are substantial questions if naming the system "CCKs" is correct.Mammals have two structurally related peptide hormones, cholecystokinin and gastrin, which can be further processed to smaller peptides of variable lengths. For these peptides to be active they need to have their C-terminal sequence GWMDFamide intact, which is, therefore, the hallmark of these gastrin/CCK peptide hormones. Also other chordates like amphibians and the tunicate *Ciona intestinalis* produce peptides (caerulein, respectively cionin) having the C-terminal GWMDFamide sequence. In addition to this C-terminal GWMDFamide hallmark, all chordate neuropeptides have a sulfated tyrosin residue located either one or two residues before the GWMDFamide sequence. Surprisingly, the GWMDFamide hallmark is absent in the two starfish peptides (none of these residues are there at corresponding positions), although a sulfated tyrosine is present at the correct position. The same situation is found in protostome invertebrates, such as insects, mollusks, and annelids. In these protostomes, however, the peptides are correctly named sulfakinins, due to a sulfated tyrosine residue located seven residues away from the C-terminus. Because the starfish peptides lack the GWMDFamide hallmark and, thus, much more resemble the protostome sulfakinins than mammalian CCKs, there is no justification to name them CCK's in the manuscrpt. Instead, they should be called sulfakinins. Sulfakinin is also a more appropriate name, because it refers to a structure, while CCK refers to a physiological action, namely the emptying of the gall bladder, which, of course, is not relevant in a starfish. Calling the starfish peptides CCK's is also leading to confusions, because it insinuates that the peptides would act like CCK and this is what is actually happening in the title of the manuscript: "Ancient role of cholecystokinin-type signalling in inhibitory regulation of feeding processes revealed in an echinoderm". Figures 6 and 7 of the manuscript show that living starfish contract their stomach after injection of the Arub -CCK's. However, also isolated muscle preparations that don't have any relation to feeding, such as tube foot preparations do contract after the addition of Arub -CCKs (Figure 6). This tells the reader of the manuscript that the Arub -CCK's are myoexcitatory, or in other words that they are"kinins", just like numerous other invertebrate and vertebrate neuropeptides are kinins. However, in the Discussion (line 824) the authors of this manuscript argue that the Arub -CCK's act like a human CCK. Their reasoning: Arub -CCK's in starfish fulfill the role of mammalian CCK, because both peptides are myoexitatory. But these arguments are, of course, not acceptable. In mammals , substance P, neuromedin, oxytocin, endothelin, acetylcholine, prostaglandin and many other neurohormones and transmitters activate smooth muscles. In insects about half of all isolated neuropeptides stimulate muscle contraction. Myoexcitation is a very unspecific property of neuropeptides and cannot be used as a criterium to establish a functional or evolutionary link or an evolutionarily conserved property between peptides. The conclusion that Arub-CCKs are the functional equivalents of human CCK, therefore, is not supported by the experiments from Figures 6 and 7. These experiments only show that the Arub-CCKs are kinins, which stimulate smooth muscle cells and inhibit feeding.Therefore the following points are requested:– re-name these peptides sulfakinins (Arub-SKs) or at least Arub-SKs/CCKs.– Both in the Title and in the Discussion, the authors should omit suggesting a functional identity between Arub-CCKs and human CCK, because this leads to misunderstandings.

Thank you for this helpful feedback on the nomenclature of the starfish CCK/SK-type neuropeptides. We agree that the names ArCCKP, ArCCK1, ArCCK2 and ArCCKR are potentially misleading if interpreted as being indicative of greater sequence similarity with vertebrate CCK/gastrin-type peptides than with protostome SK-type peptides. Therefore, we have renamed the precursor, peptides and receptor as ArCCK/SKP, ArCCK/SK1, ArCCK/SK2 and ArCCK/SKR, which accurately reflects the sequence similarity that the starfish precursor/peptides/receptor share with vertebrate CCK/gastrin-type precursors/peptides/receptors and protostome SK-type precursors/peptides/receptors, respectively. This change in nomenclature has been implemented throughout the text and in all the figures.

Accordingly, the title of the paper has been changed from:

“Ancient role of cholecystokinin-type signalling in inhibitory regulation of feeding processes revealed in an echinoderm”

To

“Ancient role of cholecystokinin/sulfakinin-type signalling in inhibitory regulation of feeding processes revealed in an echinoderm”

This change in the title makes it clear that we are not proposing that the starfish CCK/SK-type neuropeptides are functionally more closely related to CCK in humans or vertebrates than to sulfakinins (SKs) in protostomes. Sulfakinins have been shown to exert inhibitory effects on feeding behaviour in insects. Therefore, we (and others) have concluded that inhibitory regulation of feeding processes is an evolutionarily ancient role of CCK/SK-type neuropeptides that can be traced back to the common ancestor of the Bilateria. Our study provides further evidence of this for the first time in an echinoderm and in the context of a pentaradially symmetrical animal that exhibits extra-oral feeding behaviour. We agree that myoexcitatory effects of neuropeptides in vitro alone are not sufficient to infer evolutionary conservation of a physiological/behavioural role for a neuropeptide family. However, we have obtained in vivo behavioural evidence that CCK/SK-type peptides regulate feeding behaviour in starfish. Firstly, we show that CCK/SK-type peptides trigger cardiac stomach retraction in starfish, mimicking the natural process of stomach retraction that occurs when starfish terminate feeding on prey (new Figure 8). Secondly, we show that CCK/SK-type peptides have inhibitory effects on the initiation of feeding behaviour in starfish (new Figure 9). It is based on these behavioural effects that we conclude that CCK/SK-type peptides are involved in inhibitory regulation of feeding processes in starfish.

In response to this feedback we have also changed the Impact Statement that we have provided for this paper:

*Old Impact statement*: Evolutionarily ancient role of the “satiety hormone” cholecystokinin is discovered in starfish, marine invertebrates that evert their stomach out of their mouth to feed on prey.

*New Impact statement*: Starfish feed by everting their stomach out of their mouth over prey and, interestingly, this unusual feeding mechanism is inhibited by substances similar to hormones that regulate feeding in humans.

4. Resolutions of Figure 4 and 5. Make improved Figures 4 and 5, with a higher resolution. Alternatively, for Figure 5, the authors may consider improving their staining technique, for example by considering other fixation/permeabilization methods and by applying fluorescent secondary antibodies.As the RNA localization also serves as a control that the antibody against Arub-CCK1 is likely specific, a more direct comparative juxtaposition of the whole-mount in situ images vs. antibody staining should be considered.

The poor resolution is, unfortunately, a consequence of embedding these figures within a single pdf document, as recommended for submissions to *eLife*. The original tif files for these figures were prepared at a resolution of 600 pixels/inch but unfortunately it appears that these were not made available to reviewers with the original submission, for which we apologise. With the revised submission, we have uploaded all the figures for the paper in their original high-resolution formats.

Regarding presentation of the in situ hybridisation data and the immunohistochemistry, we did consider presenting these side-by-side. However, we decided not to because there would be a mismatch in the number of figure panels; this is because immunohistochemistry reveals the expression of ArCCK/SK1 in the axons of motoneurons, whereas mRNA in situ hybridisation only reveals expression in the hyponeural cell bodies of motoneurons. This mismatch does not apply to neuropeptides that are only expressed in ectoneural neurons (e.g. Asterotocin – see Odekunle et al., 2019 *BMC Biology*).

5. Please justify why a lot of gaps are introduced in the alignment of Figure 1. Gaps are often introduced to suggest (often in a biased way) structural similarities. For peptide ligands of GPCRs, the length of the peptides or the distance between two residues are often of crucial importance for binding to the receptor. The authors should also consider to provide an alignment with no gaps, which gives a better impression of structural relationships.

Inclusion of gaps is often necessary to optimally align orthologous neuropeptides from different taxa because lineage specific deletions/insertions of amino acids are a common feature of neuropeptide evolution. Accordingly, previously publications that have compared the sequences of CCK/SK-type peptides in different bilaterian taxa have included gaps in the alignments (see, for example, Figure 3B of Schwartz et al., 2018; *Sci. Rep*. 8, 16424). An alignment of CCK/SK-type peptides from different bilaterian phyla without gaps would not be informative in showing those residues that are conserved between taxa.

6. The Discussion is very verbose and unnecessarily narrative. Reduce it by 33% (from 9 to 6 manuscript pages) to make it more readable. For instance, the authors write that the DYG motiv seen in the Arub-CCKs is occurring in many other CCK's from other animal taxa (line 669). However, it is not present in the canonical human CCK's, gastrins, frog caerulein, and tunicate cionin. Therefore, omit this part of the Discussion.

We agree that the discussion was quite long and to reduce its length we have removed sections of text where we made comparisons with the expression and actions of other neuropeptides that have been functionally characterised in *A. rubens*, which is not essential material for this paper with its focus on CCK/SK-type neuropeptides. With removal of this text, the length of the manuscript has been substantially reduced. However, because of the change of peptide/precursor/receptor nomenclature and inclusion of discussion of new receptor expression data, in response to reviewer feedback, it was not possible to reduce the length of the discussion by as much as 33%.

Regarding the DYG motif, it does appear to be an evolutionarily ancient characteristic of CCK/SK-type peptides that has been conserved in insects, lophotrochozoans and echinoderms, but lost in chordates and nematodes. Therefore, in response to reviewer feedback we have modified the discussion text (lines 460-463): “In ArCCK/SK1 the tyrosine (Y) residue is preceded by an aspartate (D) residue and followed by a glycine (G) residue; accordingly, a DYG motif is also a feature of many CCK/SK-type peptides in lophotrochozoans and arthropods, suggesting that this may be an ancestral characteristic that has been lost in some bilaterian taxa (e.g. chordates, nematodes).

[Editors' note: further revisions were suggested prior to acceptance, as described below.]The manuscript has been improved significantly, however there is one remaining issue that we would ask you to address, as outlined below:Reviewer 3 brought up the aspect of re-naming again (for detailed argumentation see their comment). In essence the request is to change the name from CCK/SKP to SK/CCK, as this request then also gained support in the discussion among reviewers/RE. We acknowledge that the name CCK had already been used in 2016, but think that the recognition won't be hampered by switching the order.

We have changed the peptide/precursor/receptor names used from CCK/SK to SK/CCK throughout the manuscript and we have changed the title of the paper to “Ancient role of sulfakinin/ cholecystokinin-type signalling in inhibitory regulation of feeding processes revealed in an echinoderm.